# CD74 is a regulator of hematopoietic stem cell maintenance

**Shirly Becker-Herman[1], Milena Rozenberg[1], Carmit Hillel-Karniel[1], Naama Gil-Yarom[1], Mattias P. Kramer[1], Avital Barak[1], Lital Sever[1], Keren David** **[1], Lihi Radomir[1], Hadas Lewinsky[1], Michal Levi** ⓘ**[1], Gilgi Friedlander[2], Richard Bucala[3], Amnon Peled[4], Idit Shachar** ⓘ**[1]** *

**1** Department of Immunology, Weizmann Institute of Science, Rehovot, Israel, **2** Ilana and Pascal Mantoux Institute for Bioinformatics and Nancy and Stephen Grand Israel National Center for Personalized Medicine, Weizmann Institute of Science, Rehovot, Israel, **3** Internal Medicine, Yale School of Medicine, New Haven, Connecticut, United States of America, **4** Hadassah Hebrew University Hospital, Goldyne Savad Institute of Gene Therapy, Jerusalem, Israel

* idit.shachar@weizmann.ac.il

**Data Availability Statement:** All raw data for the Figures presented in the manuscript can be found in the Supplementary Information. The RNA-Seq data discussed in this publication have been deposited in NCBI's Gene Expression Omnibus and

## Abstract

Hematopoietic stem and progenitor cells (HSPCs) are a small population of undifferentiated cells that have the capacity for self-renewal and differentiate into all blood cell lineages. These cells are the most useful cells for clinical transplantations and for regenerative medicine. So far, it has not been possible to expand adult hematopoietic stem cells (HSCs) without losing their self-renewal properties. CD74 is a cell surface receptor for the cytokine macrophage migration inhibitory factor (MIF), and its mRNA is known to be expressed in HSCs. Here, we demonstrate that mice lacking CD74 exhibit an accumulation of HSCs in the bone marrow (BM) due to their increased potential to repopulate and compete for BM niches. Our results suggest that CD74 regulates the maintenance of the HSCs and CD18 expression. Its absence leads to induced survival of these cells and accumulation of quiescent and proliferating cells. Furthermore, in in vitro experiments, blocking of CD74 elevated the numbers of HSPCs. Thus, we suggest that blocking CD74 could lead to improved clinical insight into BM transplant protocols, enabling improved engraftment.

## Introduction

Host immunity requires a constant renewal of red blood cells and leukocytes throughout life, as these cells have a restricted life span. Hematopoietic cell turnover is enhanced following acute stress situations, such as infections or irradiation, by the proliferation of hematopoietic stem cells (HSCs) and progenitor cells (HPCs), which respond to these conditions. The hematopoietic stem and progenitor cells (HSPCs) are a small population of undifferentiated cells that reside in the bone marrow (BM). HSCs are defined by their capacity for self-renewal and ability to differentiate into all blood cell lineages. Another distinct feature of these cells is their ability to migrate out of the BM to the peripheral blood. This process is enhanced under stress as a part of the host mechanisms of defense and repair. In addition, HSCs injected to the

are accessible through GEO Series accession number GSE163661 (https://www.ncbi.nlm.nih.gov/geo/query/acc.cgi?acc= GSE163661). The FCS files for the flow cytometry data can be found in the FlowRepository (https://flowrepository.org/) with accession number FR-FCM-Z3F2.

**Funding:** I.S. was supported by the financial support of the Binational Science Foundation (BSF) grant no 711979 and The Helen and Martin Kimmel Stem Cell Research Institute https://centers.weizmann.ac.il/kimmel-stem-cell/. Both funders had no role in study design, data collection and analysis, decision to publish, or preparation of the manuscript.

**Competing interests:** The authors have declared that no competing interests exist.

**Abbreviations:** 5-FU, 5-fluorouracil; BM, bone marrow; BrdU, bromodeoxyuridine; CB, cord blood; CFU-C, colony-forming unit cell; ChIP-seq analysis, chromatin immunoprecipitation-sequencing; FACS, fluorescence-activated cell sorting; HIF, hypoxia-induced factor; HPC, hematopoietic progenitor cell; HSC, hematopoietic stem cell; HSPC, hematopoietic stem and progenitor cell; MIF, migration inhibitory factor; NAC, N-acetyl-L-cysteine; PB, peripheral blood; RNA-seq, RNA sequencing; ROS, reactive oxygen species; SLAM, signaling lymphocytic activation molecule; WT, wild-type.

blood stream, as performed in BM transplantation, can home to the BM and reestablish the HSC pool as a lifelong reservoir of new blood and immune cells [1].

The BM is the main site of adult hematopoiesis, and the majority of HSPCs remain confined to the BM microenvironment in a quiescent nonmotile state maintained via adhesive interactions [2–4]. Under stress conditions, undifferentiated progenitor cells can be triggered by their microenvironment to undergo enhanced proliferation and differentiation, to address the demand of the immune and hematopoietic systems for new leukocytes and blood cells.

Cells in the BM microenvironment maintain a functioning pool of precursor cells regulated by cytokines, by chemokines, and by additional lipid effectors. The chemokine CXCL12 and its primary receptor CXCR4 are essential for adhesion and retention of HSPCs in the mouse BM [5,6]. During homeostasis in the steady-state, CXCR4 is expressed by hematopoietic cells in addition to stromal cells, which are the main source for CXCL12 in the BM. CXCR4$^+$ HSPCs tightly adhere to BM stromal cells, which express functional, membrane-bound CXCL12 [6]. The CXCL12/CXCR4 pathway is involved in regulation of migration, survival, and development of human hematopoietic cells. Increased expression levels of CXCL12 and CXCR4 induce proliferation of hematopoietic progenitors and recruitment of bone-resorbing osteoclasts, osteoblasts, neutrophils, and other myeloid cells, leading to leukocyte mobilization [7]. In addition, reduced CXCR4 expression levels might result in an amplified mobilization response and cell proliferation in the BM [8].

CD74 mRNA is expressed in HSPCs [9–11]; however, the role of this receptor in HSPCs was never analyzed. CD74 is a type II integral membrane protein that is expressed many cell types. The CD74 chain was initially described to function mainly intracellularly as an MHC class II chaperone [12]. A small proportion of CD74 is modified by the addition of chondroitin sulfate (CD74-CS), and this form of CD74 is expressed on the cell surface. It was previously shown that macrophage migration inhibitory factor (MIF) binds to the CD74 extracellular domain, a process that results in the initiation of a signaling pathway [13].

MIF is also a noncognate ligand of the CXCRs, CXCR2, and CXCR4, and biochemical evidence suggests that these chemokine receptors could act as additional signal-transducing CD74 coreceptors upon MIF stimulation [14,15]. Importantly, structural and functional interactions between CD74 and the MIF chemokine receptor, CXCR4, have been proposed [14,16].

Our previous studies have shown that CD74 expressed on healthy and malignant B cells is directly involved in regulating murine mature B cell survival [17–20] through a pathway leading to the activation of transcription mediated by the NF-κB p65/RelA homodimer and its coactivator, TAFII105 [21]. NF-κB activation is mediated by the cytosolic region of CD74 (CD74-ICD), which is liberated from the membrane, and translocates to the nucleus [22]. Moreover, we demonstrated that CD74 stimulation by MIF enables augmented expression of antiapoptotic proteins in a CD44-dependent manner. In addition, we recently characterized the transcriptional activity of CD74-ICD. We showed that following CD74 activation, CD74-ICD interacts with the transcription factors RUNX and NF-κB and binds to proximal and distal regulatory sites enriched for genes involved in apoptosis, immune response, and cell migration. This leads to regulation of expression of these genes.

In the current study, the role of the MIF/CD74 axis in HSPCs was followed. We show that CD74 plays a crucial role in HSPC maintenance. Deficiency of CD74 and MIF leads to enhanced survival and accumulation of HSPCs in the BM. The enlarged pool of HSCs give rise to higher numbers of HSPCs and the various immune cell lineages. Cells lacking CD74 demonstrated an advantage in repopulating the host environment, as seen by the significantly higher levels of those cells when compared to the wild-type (WT) cells in mixed chimera. Thus, our study could lead to improved clinical insight into factors governing the efficacy of BM transplantation protocols, as well as diseases associated with hematopoietic failure.

## Results

### Expansion of hematopoietic stem and progenitor cell populations in CD74$^{-/-}$ mice

CD74 mRNA levels were analyzed in HSPCs (Lin-Sca-1+c-Kit+; LSK) [9–11], which are enriched for hematopoietic stem (CD34-Lin-Sca-1+c-Kit+; HSCs) and progenitor (CD34 + Lin-Sca-1+c-Kit+; HPCs) cells (gating is shown in S1A–S1D Fig) [10,23,24]. To determine the role of CD74 in these cells, we first confirmed CD74 expression by analyzing its mRNA levels in sorted HSC populations. As shown in S1E Fig, CD74 message was detected in WT stem cells and was absent from the CD74-deficient (CD74$^{-/-}$) cells. We next analyzed CD74 cell surface protein levels by fluorescence-activated cell sorting (FACS). As shown in Fig 1A, CD74 was expressed on the surface of these populations in low levels, similar to those expressed on mature B cells [12]. To determine CD74 function in these cells, cell numbers of the various stem and progenitor populations in WT and CD74$^{-/-}$ mice were compared. While similar

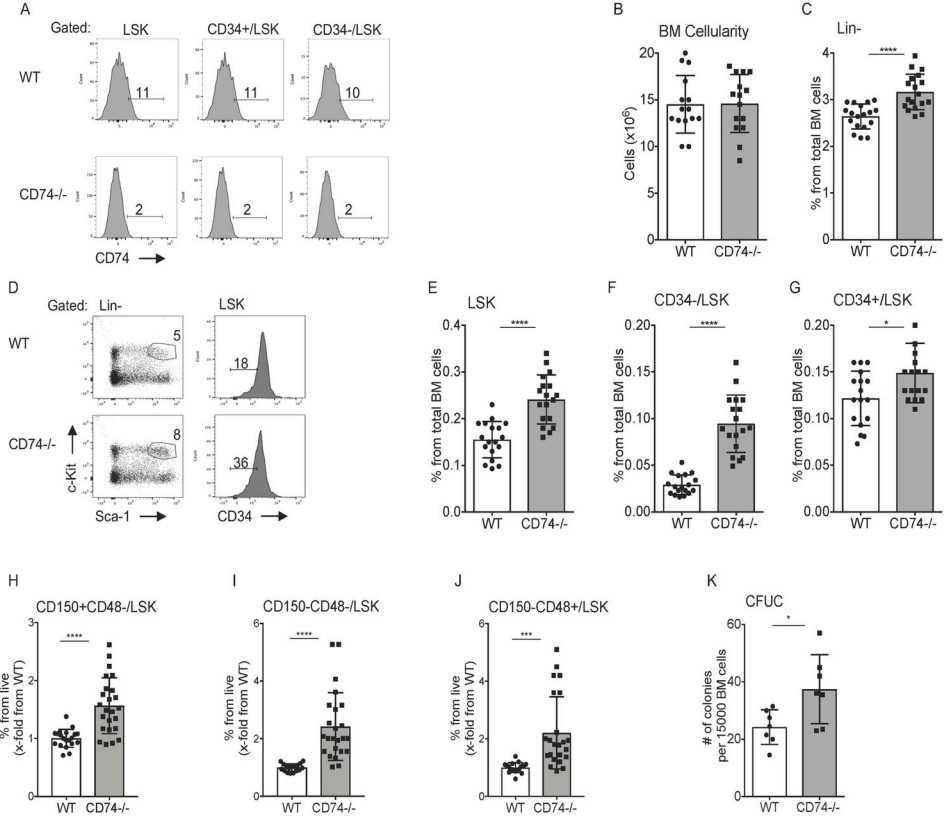

**Fig 1. Expansion of HSPCs in the BM of CD74$^{-/-}$ mice. (A)** BM cells derived from WT or CD74$^{-/-}$ were purified. Histograms show representative analysis of CD74 expression on HSPCs in WT and CD74$^{-/-}$ mice. $n$ = 3. **(B)** Total BM cellularity per femur and tibia in WT and CD74$^{-/-}$ mice, Data A in S1 Data. **(C–J)** The percent of the different populations in WT and CD74$^{-/-}$-derived BM cells. **(C)** Lin-; Data B in S1 Data **(D)** Representative FACS analysis of WT and CD74$^{-/-}$ HSPCs; (E) LSK; Data C in S1 Data **(F)** CD34-/LSK; Data D in S1 Data and **(G)** CD34+; Data E in S1 Data **(H)** CD150+CD48-LSK; Data F in S1 Data **(I)** CD150-CD48-/LSK; Data G in S1 Data and **(J)** CD150-CD48 +/LSK; $n$ = 14–18, Data H in S1 Data. **(K)** CFUC assay: Total BM cells from WT and CD74$^{-/-}$ mice were seeded at 15,000 cells/mL in semisolid cultures supplemented with cytokines and nutrients. CFU-C were counted 7 days later; $n$ = 7, Data I in S1 Data. Bars show SEM. Unpaired two-tailed $t$ test $^*p < 0.05$; $^{**}p < 0.01$; $^{***}p < 0.001$; $^{****}p < 0.0001$. The fcs files and gates can be found at the Flow Repository (accession number FR-FCM-Z3F2). BM, bone marrow; CFU-C, colony-forming unit cell; FACS, fluorescence-activated cell sorting; HSPC, hematopoietic stem and progenitor cell; WT, wild-type.

numbers of BM cells were detected in WT and CD74$^{-/-}$ mice (Fig 1B), a significant increase in the Lineage marker-negative (CD11b-,Gr-1-,CD3-,B220-and Ter117-; Fig 1C) and LSK (Fig 1D and 1E) populations was observed in the CD74$^{-/-}$ animals. Moreover, an elevation was also detected in the early lymphoid-committed precursors (cKit-Sca1+) and common myeloid progenitor populations (cKit+Sca1-) (S1F Fig). Next, the percent of HSCs and HPCs was compared. As shown in Fig 1F and 1G, a significant increase in both the CD34$^+$ and CD34$^-$ populations was detected in mice lacking CD74, with a more significant elevation of the HSC CD34$^-$ population. This accumulation was detected as well in CD74$^{-/-}$ progeny of newly crossed WT and CD74$^{-/-}$ mice, relative to CD74 expressing littermates (S1G Fig).

These progenitor populations are also characterized by expression of the signaling lymphocytic activation molecule (SLAM) family members, CD150 and CD48 [10,23,24]. Similarly, we observed an increase in the percent of HSCs and progenitors in CD74$^{-/-}$ mice when identified by FACS analysis for CD150, CD48, Lin$^-$, ckit$^+$ and Sca-1$^+$ (Fig 1H–1J). These results suggest elevated stem and progenitor cell populations in mice lacking CD74. Since CD74 regulates the expression of SLAM receptors [25], we decided to focus on the CD34 marker for stem cell analysis in this study.

To examine the in vitro repopulation potential of CD74-deficient stem cells, the ability of HSPCs to proliferate and differentiate into colonies in vitro was analyzed by a colony-forming unit cell assay (CFU-C assay). As seen in Fig 1K, and as previously demonstrated [26], higher numbers of colonies were generated from CD74$^{-/-}$ BM when compared to WT cells.

Next, the expression of the CD74 ligand, MIF, and its coreceptor, CD44, were analyzed. Similar intracellular MIF (S1H and S1I Fig) and cell surface CD44 (S1J and S1K Fig) expression levels were detected in CD74-deficient HSPCs compared to WT. To directly determine the role of MIF in HSPC accumulation, total BM cells were extracted from WT and MIF$^{-/-}$ mice, and progenitor populations were analyzed. Higher numbers of Lin neg and HSC populations were observed in the MIF$^{-/-}$ mice compared to WT animals (S2A–S2D Fig). However, the differences were not as pronounced as in the CD74-deficient mice. This could be explained by partial compensation by the MIF homologue, MIF2 [27]. These results suggest that MIF and its receptor, CD74, limit HSPC number.

## CD74$^{-/-}$ HSPCs demonstrate enhanced long-term self-renewal capacity

To determine whether the expansion of HSPCs in CD74$^{-/-}$ mice results from an effect intrinsic to the cells themselves, or whether the differences are due to extrinsic environmental factors, chimeric mice were generated. Total BM cells from WT or CD74$^{-/-}$ mice were transplanted into lethally irradiated WT or CD74$^{-/-}$ recipients. The animals were killed after 16 weeks, and their HSPCs were analyzed. As seen in Fig 2A–2C, elevation in Lin, LSK, and CD34$^-$ populations was detected in mice transplanted with CD74$^{-/-}$ BM compared to WT donors. Thus, the lack of CD74 in the donor cells rather than in the microenvironment contributed to HSPC accumulation. Taken together, these results indicate that the lack of MIF/CD74 signaling results in an intrinsic increase in the HSPC population in the BM.

To further evaluate the effect of the stroma and cytokines in the cellular microenvironment on CD74$^{-/-}$ HSPC numbers, BM cells were cultured in vitro in the presence of the stroma M210B4 cell line or various cytokines, and the proportion of the HSPCs out of live cells was compared. These conditions had only a minor or no effect on the fold of increase in WT and CD74$^{-/-}$ stem cell numbers (Fig 2D–2F), further demonstrating that the accumulation potential of HSPCs in the absence of CD74 is intrinsic to the stem cells.

Next, we followed the in vivo potential of CD74-negative HSPCs to repopulate and compete with WT cells. To this end, WT (CD45.1) BM cells were transplanted at a 1:1 ratio with either

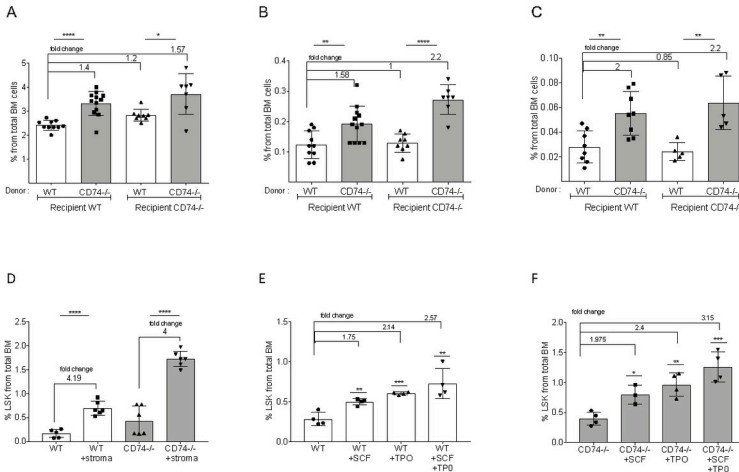

**Fig 2. CD74$^{-/-}$ HSPC expansion is cell intrinsic.** Lethally irradiated WT or CD74$^{-/-}$ mice were transplanted with either WT or CD74$^{-/-}$ total BM cells. Long-term reconstitution was evaluated 16 weeks posttransplantation. Percent of total BM cells was calculated for **(A)** LIN-; Data A in S2 Data **(B)** LSK; Data B in S2 Data and **(C)** CD34-/LSK; Data C in S2 Data $n = 5$–12. Bars show SEM. Unpaired two-tailed $t$ test $^* < 0.05$ $^{**} < 0.01$ $^{***} < 0.001$ $^{****} < 0.0001$. **(D)** WT and CD74$^{-/-}$ BM ($2*10^6$ cells) were incubated with or without $10^5$ stroma cells. After 48 h, percent LSK from live cells was analyzed; $n = 5$–6, Data D in S2 Data. **(E, F)** WT and CD74$^{-/-}$ BM ($2*10^6$) were incubated with or without 50 μg SCF and 50 μg TPO, or both SCF and TPO (50 μg each). After 48 h, percent LSK from live cells was analyzed; $n = 4$, Data E and F in S2 Data. BM, bone marrow; HSPC, hematopoietic stem and progenitor cell; SCF, stem cell factor; TPO, thrombopoietin; WT, wild-type.

CD74$^{-/-}$ (CD45.2) or WT (CD45.2) cells into lethally irradiated recipient mice (CD45.1). BM and peripheral blood (PB) populations of the mixed chimeras were analyzed at 6, 16, and 24 weeks after transplantation. While WT CD45.1 and CD45.2 chimera maintained a 1:1 ratio (S3A–S3H Fig), CD74$^{-/-}$-derived BM cells exhibited a growth advantage over the WT populations (Fig 3 and S3I–S3L Fig). The dramatic overgrowth of CD74$^{-/-}$ cells was observed as early as 6 weeks posttransplant and was maintained throughout the experiment. A significant advantage of CD74$^{-/-}$ total BM cells (Fig 3A and 3B), myeloid (Fig 3C), total B cells (Fig 3D), and HSPC populations (Fig 3E and 3F) was observed at various time points tested. In the B cell lineage, the growth advantage of the CD74$^{-/-}$ population was detected from early stages of B cell differentiation through the formation of immature B cells in the BM (Fig 3G). This advantage disappeared at the mature stage (Fig 3H) due to the separate role of CD74 as a survival receptor on these cells [28]. Analysis of PB populations revealed an advantage similar to that observed in the BM. The myeloid (S3I Fig) immature B (S3J Fig), and LSK (S3K Fig) populations accumulated, while the mature B cell population (S3L Fig) was reduced in the absence of CD74. We could not follow T cell populations in these chimeric mice due to the difficulty of completely eliminating host T populations by irradiation.

To determine whether this accumulation results from differential homing of the cells or their improved quality, the homing of HSPCs to the BM was followed up to 1 week after BM transplantation. WT and CD74$^{-/-}$ HSPCs showed similar homing potential to the BM, and no differences in the number of WT versus CD74$^{-/-}$ HSPCs in the BM were detected during the first week after transplantation (S4A–S4E Fig). This suggests that the accumulation of CD74-deficient HSPCs does not result from their induced homing, but rather from their enhanced potential to repopulate the BM compartment. Thus, the CD74-deficient BM cells have an intrinsic advantage in repopulation and accumulation in the BM niche.

Next, to directly test the capacity of CD74$^{-/-}$ HSPCs to repopulate the immune system, sorted LSK populations from WT (CD45.1) and CD74$^{-/-}$ (CD45.2) mice were transplanted at

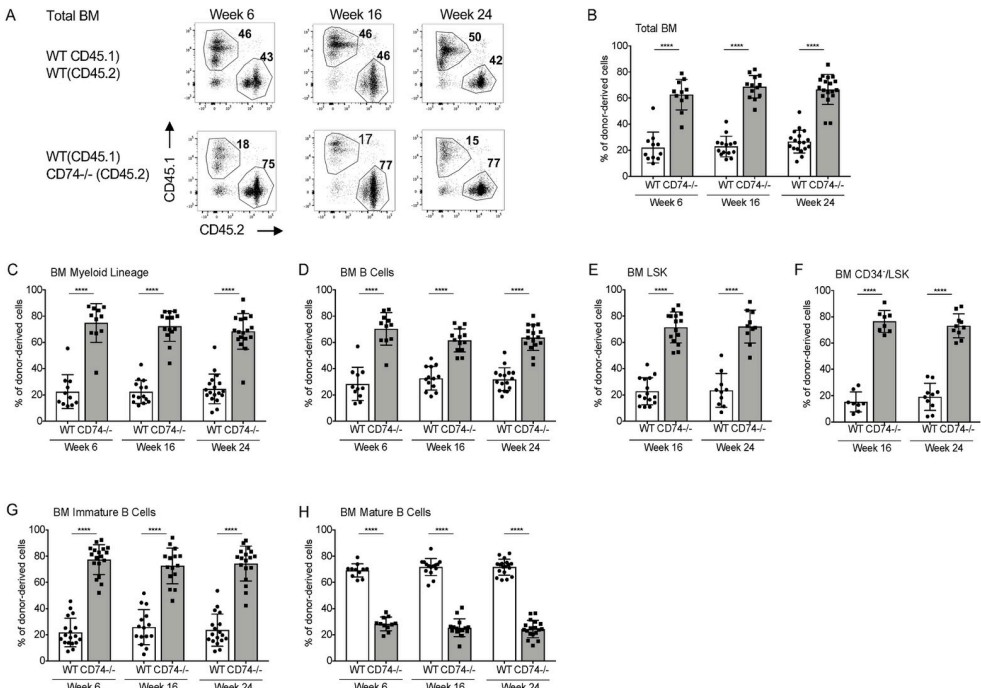

**Fig 3. CD74$^{-/-}$ HSPCs have an advantage in BM repopulation.** Lethally irradiated WT (CD45.1) mice were transplanted with BM derived from WT (CD45.1) and WT (CD45.2) at a 1:1 ratio, or BM derived from WT (CD45.1) and CD74$^{-/-}$ (CD45.2) mice at a 1:1 ratio. **(A)** Representative BM FACS staining. Percent of donor-derived cells was analyzed in the BM after 6, 16, and 24 weeks in **(B)** Total BM cells; Data A in S3 Data **(C)** myeloid cells (CD11B+); Data B in S3 Data **(D)** B cells (B220+); Data C in S3 Data **(E)** LSK; Data D in S3 Data **(F)** CD34-/LSK; Data E in S3 Data **(G)** immature BM B cells (B220+IgD-); Data F in S3 Data **(H)** mature BM B cells (B220+ IgM+ IgD+); Data G in S3 Data. $n = 8$–18. Bars show SEM. Unpaired two-tailed $t$ test $^*p < 0.05$; $^{**}p < 0.01$; $^{***}p < 0.001$; $^{****}p < 0.0001$. The fcs files and gates can be found at the Flow Repository (accession number FR-FCM-Z3F2). BM, bone marrow; FACS, fluorescence-activated cell sorting; HSPC, hematopoietic stem and progenitor cell; WT, wild-type.

a 1:1 ratio to irradiated recipient mice (CD45.1) to generate competitive BM chimera. BM populations of the mixed chimeras were analyzed 6 and 18 weeks following the transplantation. As shown in Fig 4A–4E, a significant advantage of CD74$^{-/-}$ BM cells was observed at both 6 and 18 weeks postengraftment. These results indicated that LSK lacking CD74 are more efficient in repopulating the host environment, as seen by the significantly higher levels of those cells when compared to the WT (CD45.1).

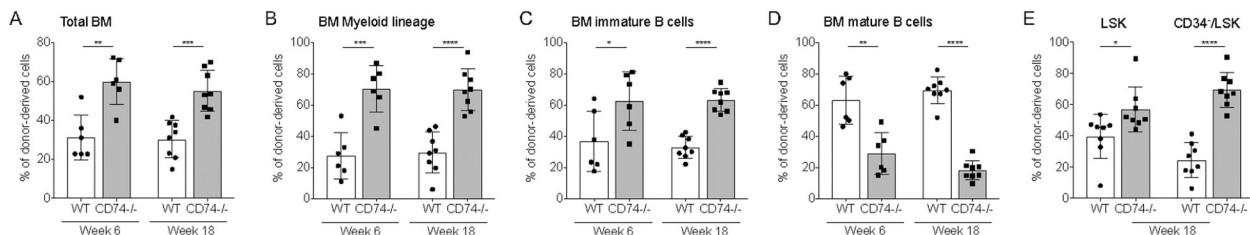

**Fig 4. CD74$^{-/-}$ HSPCs show a higher potential to repopulate the BM.** Lethally irradiated WT CD45.1 recipient mice were reconstituted with $7.5^*10^4$ sorted LSK cells from WT (CD45.1), and $7.5 \times 10^4$ sorted LSK from CD74$^{-/-}$ (CD45.2) at a 1:1 ratio. Percent of donor-derived cells was analyzed in the BM after 6 and 18 weeks. **(A)** Total BM cells; Data A in S4 Data **(B)** myeloid cells; Data B in S4 Data **(C)** immature BM B cells; Data C in S4 Data and **(D)** mature BM B cells, Data D in S4 Data. **(E)** Percent of donor-derived cells was analyzed in LSK and CD34-LSK cells 18 weeks posttransplant. $n = 6$–8. Data E in S4 Data. $n = 6$–8. Bars show SEM. Unpaired two-tailed $t$ test $^*p < 0.05$; $^{**}p < 0.01$; $^{***}p < 0.001$; $^{****}p < 0.0001$. BM, bone marrow; HSPC, hematopoietic stem and progenitor cell; WT, wild-type.

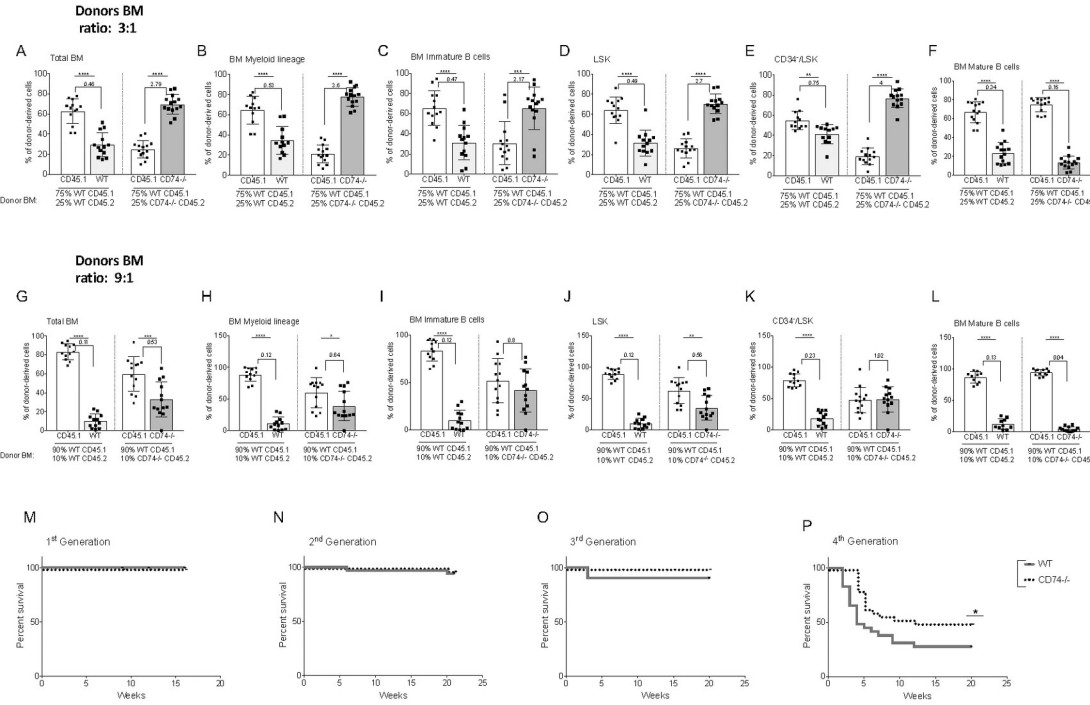

**Fig 5. CD74$^{-/-}$ HSPCs demonstrate enhanced long-term self-renewal capacity. (A–F)** Lethally irradiated WT(CD45.1) mice were transplanted with BM derived from WT (CD45.1) and WT(CD45.2) at a 3:1 ratio, or BM derived from WT (CD45.1) and CD74$^{-/-}$ (CD45.2) mice at a 3:1 ratio. Mice were analyzed 16 weeks after transplantation. Graphs show percent of donor derived cells from both WTCD45.1/WTCD45.2 and WTCD45.1/CD74$^{-/-}$ CD45.2 chimera. (A) Total BM cells; Data A in S5 Data (B) BM myeloid cells; Data B in S5 Data (C) BM immature B cells; Data C in S5 Data (D) LSK; Data D in S5 Data (E) CD34-/LSK; Data E in S5 Data (F) Mature BM B cells; Data F in S5 Data. $n = 13$. **(G–L)** Lethally irradiated WT(CD45.1) mice were transplanted with BM derived from WT (CD45.1) and WT(CD45.2) at a 9:1 ratio, or BM derived from WT (CD45.1) and CD74$^{-/-}$ (CD45.2) mice at a 9:1 ratio. Mice were analyzed 16 weeks after transplantation. Graphs show percent of donor-derived cells from both WTCD45.1/WTCD45.2 and WTCD45.1/CD74$^{-/-}$ CD45.2 chimera. (G) Total BM cells; Data G in S5 Data (H) BM myeloid cells; Data H in S5 Data (I) BM immature B cells; Data I in S5 Data (J) LSK; Data J in S5 Data (K) CD34-/LSK; Data K in S5 Data (L) Mature BM B cells; Data L in S5 Data. $n = 12$. Bars show SEM. Unpaired two-tailed $t$ test $^*p < 0.05$; $^{**}p < 0.01$; $^{***}p < 0.001$; $^{****}p < 0.0001$. **(M–P)** Survival curves for serial transplantation assay. BM from 6 donors from each genotype were transplanted to 4–5 lethally irradiated hosts. After 10–12 weeks, one mouse from each donor served as a donor for the subsequent transplant. Each host were transplanted with $2 \times 10^6$ BM cells. Log-rank test $^*<0.05$, $n = 20$–30 mice in each transplant per genotype. Data M–P in S5 Data.

CD74$^{-/-}$-derived BM cells show an advantage over the WT populations in BM transplantation. The dramatic overgrowth of CD74$^{-/-}$ cells might result from their higher numbers of progenitor cells, from their higher potential to repopulate their niche, or both. To determine the basis for the enhanced ability of CD74$^{-/-}$ HSCs to repopulate the BM, we wished to follow the competition under conditions in which the HSC WT and CD74$^{-/-}$ numbers are similar. We therefore analyzed BM repopulation after serial dilutions (3:1, 7:1, and 9:1). Chimeric mice were generated at a ratio of approximately 3:1 in favor of the WT, which represents the injection of similar numbers of HSCs. As shown in Fig 5A–5F, while the WT (CD45.1):WT (CD45.2) chimeric mice preserved a ratio of 3:1, CD74$^{-/-}$-derived BM cells retained an advantage over the WT populations and resulted in a dramatic takeover of CD74$^{-/-}$ cells as early as 6 weeks posttransplant, which was maintained throughout the experiment (16 weeks). A significant numerical advantage of CD74$^{-/-}$ total BM (Fig 5A and S4F Fig) myeloid cells (Fig 5B), immature B cells (Fig 5C), LSK (Fig 5D), and CD34-/LSK (Fig 5E) was observed at this time point. The only population that could not compete was the mature B population (Fig 5F), whose survival is dependent on CD74 expression [29]. To gain further insight into the enhanced potential of CD74$^{-/-}$ HSPCs to repopulate the BM, we followed the population at

7:1 and 9:1 dilutions, as well. While in the 7:1 WT:WT chimera the proportion of CD45.1: CD45.2 remained 7:1, a similar ratio or some advantage of CD74$^{-/-}$ cells in the total BM (S4G Fig), CD11b (S4H Fig), immature B (S4I Fig), and LSK (S4J Fig) populations was observed in the WT:CD74$^{-/-}$ chimera. Interestingly, at this dilution, the CD34-/LSK WT:CD74$^{-/-}$ population still showed an advantage for the CD74$^{-/-}$ cells (S4K Fig). The only population that remained lower at this dilution were the CD74-deficient mature B cells (S4L Fig). At a 9:1 dilution ratio, the advantage of CD74$^{-/-}$ cells was diminished in all populations (Fig 5G–5L), although even at this dilution, the ratio of CD74$^{-/-}$ cells (CD45.2) to WT (CD45.1) was higher compared to the ratio of WT CD45.2 to WT CD45.1. Thus, CD74-deficient HSCs have a stronger potential to repopulate the BM and compete for the niches relative to WT HSPCs.

To evaluate the long-term self-renewal and functional properties of CD74$^{-/-}$ HSCs, a serial BM transplantation assay was performed. BM cells from 6 WT and 6 CD74$^{-/-}$ mice were isolated, and the cells were serially transplanted into lethally irradiated WT mice. Each host was transplanted with $2 \times 10^6$ BM cells. During the first 3 first cycles, no significant differences were observed between the WT and CD74$^{-/-}$ groups, with high survival rates of all mice (Fig 5M–5O). However, by the fourth transplantation cycle, CD74$^{-/-}$-transplanted mice showed a better survival rate compared to the WT mice (57% compared to 33%) (Fig 5P). Thus, the absence of CD74 in HSCs results in accumulation of cells with a higher potential to repopulate the BM.

## CD74 regulates stem cell survival

Next, we wished to identify the molecular mechanism regulating the HSPC accumulation in CD74-deficient mice. The accumulation of stem cells might result from their elevated retention in the BM niche or up-regulation of their proliferation or survival. Since CXCR4 plays a major role in retention of HSCs and HSPCs [8,30,31], we next wished to follow the role of CD74 in CXCR4 expression and function in HSPCs. We previously showed that following activation of CD74 expressed on CLL cells, CD74-ICD binds the chromatin of the CXCR4 promoter ([32]; S5A Fig). Expression of cell surface CXCR4 was therefore analyzed on HSPCs derived from WT and CD74-deficient mice. As shown in Fig 6A and 6B, a reduction in the expression of CXCR4 on the cell surface was observed on CD74$^{-/-}$ cells. This result shows that CD74 regulates CXCR4 expression. To determine whether the reduced levels of CD74 affect HSPC retention, the total counts of HSPCs in WT and CD74$^{-/-}$ PB were compared. As shown in Fig 6C–6E, a dramatic elevation in the number of HSPCs in the circulation of CD74-deficient mice was detected. However, treatment of the mice with AMD3100 induced mobilization of both WT and CD74$^{-/-}$ HSPCs to the same degree (6- to 7-fold; Fig 6F). All studies performed with CXCR4-deficient HSPCs demonstrated a requirement of this receptor for efficient engraftment. Nevertheless, transient inhibition of CXCR4 had no adverse effects on the engraftment capacity of the HSPCs [8]. CD74 deficiency partially reduces CXCR4 expression; however, our results suggest a negligible role for CXCR4 in the induced mobilization of CD74-deficient cells.

It was recently shown that blocking CXCR4 induces an increase in the cycling activity of the HSPCs [8]. We therefore analyzed the role of CD74 in HSPC proliferation. To determine whether CD74 controls cell proliferation and the cell cycle, Ki67 levels were followed. Higher numbers of both quiescent (Ki67) and cycling (Ki67$^+$) stem cells (CD34-LSK) (Fig 6G) and progenitors (CD34+LSK) (Fig 6H) were detected in CD74$^{-/-}$ mice. However, the ratio between Ki67$^-$ and Ki67$^+$ populations did not change (Fig 6I). To further follow cell proliferation in mice lacking CD74, a 5-bromodeoxyuridine (BrdU)-labeling experiment was performed. Mice were fed 0.8 mg/ml BrdU in their drinking water for 3 days, and BrdU

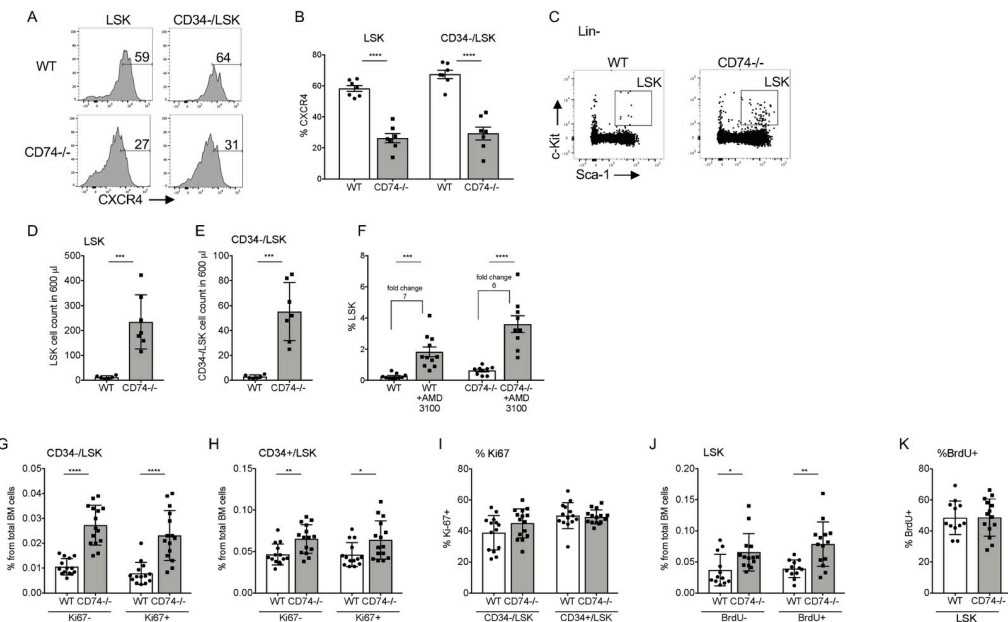

**Fig 6. Accumulation of HSPCs is not CXCR4 dependent. (A, B)** FACS analysis of CXCR4 expression on BM LSK and BM CD34-/LSK of WT and CD74$^{-/-}$ mice, $n = 7$, Data A in S6 Data. Representative histograms are shown. **(C–E)** FACS analysis for HSPCs in the PB of WT and CD74$^{-/-}$. (C) Dot plot analysis of LSK in WT and CD74$^{-/-}$ mice. **(D, E)** Cell number of (D) LSK and (E) CD34-LSK in 600 μl blood. WT $n = 6$ CD74$^{-/-}$ $n = 7$, Data B and C in S6 Data. **(F)** AMD3100 (20 mg/kg$^{-1}$) was injected to WT and CD74$^{-/-}$ mice. After 2 h, percent of LSK in the PB was analyzed; $n = 9$–11, Data D in S6 Data. **(G–I)** FACS staining of WT and CD74$^{-/-}$ HSPCs for Ki-67. Results are presented as: (G) percent of CD34-/LSK Ki-67 and CD34-/LSK Ki-67+ from total BM cells, Data E in S6 Data; (H) percent of CD34+/LSK Ki-67- and CD34+/LSK Ki-67+ from total BM cells, Data F in S6 Data; and (I) percent of Ki-67+ from CD34-/LSK and percent of Ki-67+ from CD34+LSK, $n = 15$, Data G in S6 Data. **(J, K)** Mice were fed with 0.8 mg/ml BrdU in their drinking water for 3 days, and BrdU incorporation was analyzed by FACS. Results are represented as: (J) percent of LSK BrdU- and LSK BrdU+ from total BM cells, Data H in S6 Data; (K) percent of BrdU+ in LSK; $n = 12$–14, Data I in S6 Data. Bars show SEM. Unpaired two-tailed $t$ test: $^*p < 0.05$; $^{**}p < 0.01$; $^{***}p < 0.001$; $^{****}p < 0.0001$. The fcs files and gates can be found at the Flow Repository (accession number FR-FCM-Z3F2). BM, bone marrow; BrdU, bromodeoxyuridine; FACS, fluorescence-activated cell sorting; HSPC, hematopoietic stem and progenitor cell; PB, peripheral blood; WT, wild-type.

incorporation was followed. As shown in Fig 6J and 6K, although higher numbers of both quiescent and cycling cells were detected in CD74$^{-/-}$ mice, no significant change in the ratio of these populations was observed. These results suggest that although the HSC and HSPC compartments are larger in mice lacking CD74, there is no overproliferation of any specific population, and the proportion of proliferating cells is similar in the WT and CD74$^{-/-}$ mice.

Electron transfer along the mitochondrial respiration chain induces the formation of reactive oxygen species (ROS) [33]. Emerging evidence shows that oxidative stress and, in particular, ROS content, influences stem cell migration, development, and self-renewal, as well as their progression through the cell cycle [1]. To determine whether oxidative phosphorylation is elevated in the absence of CD74 in stem cells, ROS levels were compared in HSPCs of CD74$^{-/-}$ and WT cells. As can be seen in Fig 7A, higher numbers of ROS$^{high}$ cells were detected in the CD74$^{-/-}$ HSPCs compared to the WT. However, no difference was observed in the ratio between ROS$^{high}$ and ROS$^{low}$ expressing cells (Fig 7B), suggesting that level of ROS expression in each subpopulation remains constant. To determine whether excess ROS contributes to the expansion of CD74$^{-/-}$ stem cells, ROS levels were reduced using the antioxidant, N-acetyl-L-cysteine (NAC) [34]. As shown in Fig 7C and 7D, treatment with NAC for 6 days

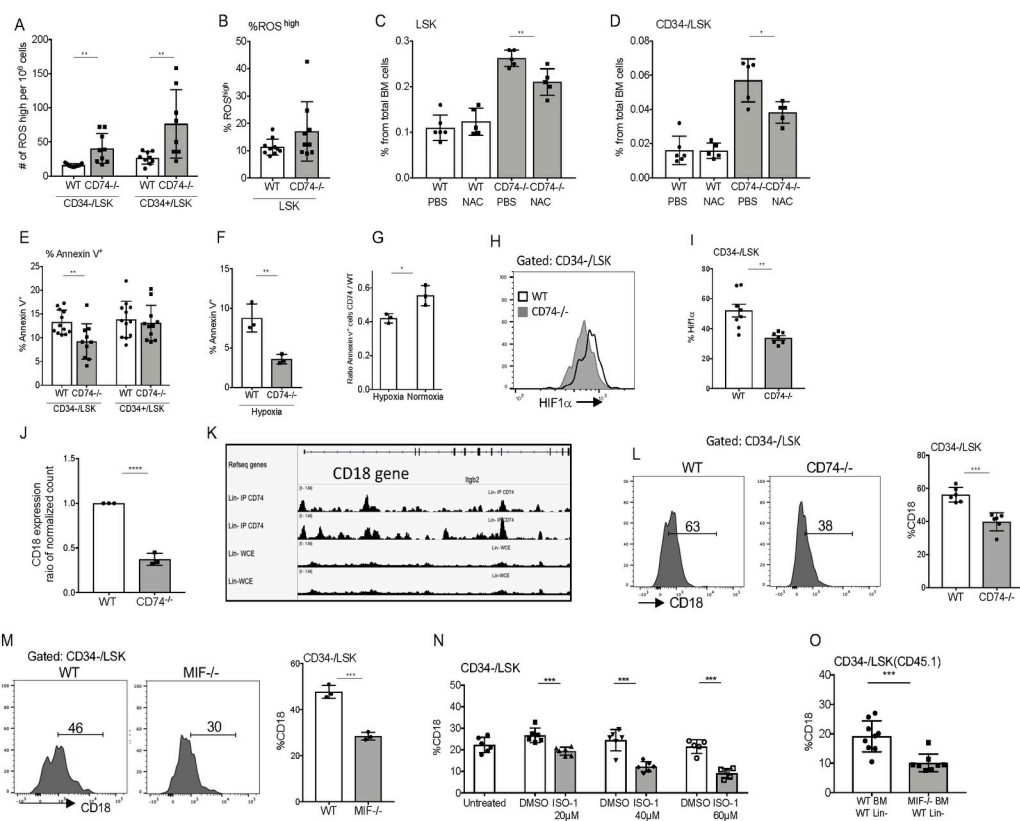

**Fig 7. CD74 regulates the survival of HSPCs and CD18 expression. (A, B)** FACS staining of WT and CD74-deficient HSPCs for ROS. (A) Results are presented as the number of ROS high cells per $10^6$ cells, $n = 9$, Data A in S7 Data. (B) Percentage of ROS$^{high}$ in LSK, Data B in S7 Data. **(C, D)** Percent of LSK (C) Data C in S7 Data, and CD34- (D) after 6 days of NAC injections (50 mg kg$^{-1}$); $n = 5$, Data D in S7 Data. **(E, F)** FACS analysis of HSPCs from WT and CD74$^{-/-}$ mice for Annexin V (E); $n = 10$–12, Data E in S7 Data, and after 24 h under hypoxic (F); $n = 3$ (each dot represents a duplicate determination), Data F in S7 Data. **(G)** Ratio of Annexin V+ CD74$^{-/-}$ to WT of HSPCs under hypoxic and normoxic conditions, Data G in S7 Data. **(H, I)** FACS analysis of HSCs from WT and CD74$^{-/-}$ mice for HIF-1α; $n = 7$–8, Data H in S7 Data. **(J)** Sorted WT and CD74$^{-/-}$ CD34-/LSK cells were analyzed for CD18 mRNA levels; $n = 3$. The bars show the DESeq2 normalized counts for the CD18 gene, Data I in S7 Data. **(K)** Binding of CD74–ICD to CD18 promoter and intron regions in Lin− samples. ChIP-seq analysis using anti-CD74 antibody. **(L)** FACS analysis of HSCs from WT and CD74$^{-/-}$ mice for CD18. Graph summarizes the results of 6 mice in each group, Data J in S7 Data. **(M)** FACS analysis of HSCs from WT and MIF$^{-/-}$ mice for CD18; $n = 3$, Data K in S7 Data. **(N)** WT and CD74$^{-/-}$ BM were cultured with or without the MIF inhibitor, ISO-1, for 48 h, percent CD18 on CD34-/LSK was analyzed by FACS; $n = 6$, Data L in S7 Data. **(O)** WT (CD45.1) Lin negative cells were cultured in the presence of WT (CD45.2) total BM or MIF$^{-/-}$ (CD45.2) total BM for 48 h. The percent CD18 on CD34-/LSK cells (CD45.1) was analyzed by FACS; $n = 8$, Data M in S7 Data. Bars show SEM. Unpaired two-tailed $t$ test *$p < 0.05$; **$p < 0.01$; ***$p < 0.001$. The fcs files and gates can be found at the Flow Repository (accession number FR-FCM-Z3F2). BM, bone marrow; ChIP-seq, chromatin immunoprecipitation-sequencing; FACS, fluorescence-activated cell sorting; HIF-1α, hypoxia-inducible factor 1 alpha; HSC, hematopoietic stem cell; HSPC, hematopoietic stem and progenitor cell; MIF, migration inhibitory factor; NAC, N-acetyl-L-cysteine; ROS, reactive oxygen species; WT, wild-type.

partially reduced the levels of HSPCs in CD74$^{-/-}$ mice. This suggests that ROS levels play a limited role in HSPC accumulation in mice lacking CD74.

To further probe the mechanism of action of CD74 in HSPCs, we wished to determine whether the higher number of CD74$^{-/-}$ HSPCs results from enhanced cell survival. Therefore, HSPCs cells were analyzed for cell survival using an Annexin V staining assay. As shown in Fig 7E, reduced apoptosis was observed in the CD74$^{-/-}$ CD34$^-$ LSK cells compared to the WT population. Thus, the higher number of CD74$^{-/-}$ stem cells might result from an increase in their survival.

Since CD74 regulates mature B cell survival, and its absence leads to cell death [35,36], we wished to understand the different, and possibly contradictory, roles of CD74 in stem cells. We suggested that hypoxic conditions that exist in the BM environment and especially the perivascular niches where the nondividing HSCs reside control the enhanced survival of CD74$^{-/-}$ cells [37]. WT and CD74$^{-/-}$ BM cells were incubated under hypoxia or with normal oxygen levels for 24 h, and cell survival was analyzed by Annexin V staining. As can be seen in Fig 7F, CD74-deficient stem cells survived better under hypoxia. Transfer of the CD74$^{-/-}$ cells to normoxic conditions reduced the advantage of these cells compared to WT stem cells (Fig 7G). These results support our suggestion that the BM hypoxic conditions play a role in the control of CD74 function.

A key regulator of adaptation processes induced by hypoxia is the transcription factor, hypoxia-induced factor (HIF), which is active only after heterodimerization of its 2 subunits α and β. Although HIF-1β is constitutively expressed, HIF-1α is induced by hypoxia. It was previously shown that MIF regulates HIF-1α expression in a CD74-dependent manner [38]. We therefore tested HIF-1α protein levels in WT and CD74$^{-/-}$ CD34- population. As shown in Fig 7H and 7I, lower levels of HIF-1α protein were detected in the CD34- populations, suggesting a role for CD74 in the control of this transcription factor expression.

To further follow CD74 regulated cascades, RNA sequencing (RNAseq) analysis of gene expression in WT and CD74$^{-/-}$ CD34-LSK cells was performed. RNAseq revealed that CD74 regulated the expression of transcription factors including KLF4, KLF2, and E2F2, and altered the expression of pathways induced by transcription factors such as IRF8, MDB2, and CEBPA, which are known to regulate HSCP maintenance (S5B Fig) [39–44]. The RNAseq also revealed a regulation of the ITGB2 and ITGB3 pathways. We further studied the *ITGB2* gene, which encodes the β2 common leukocyte subunit, CD18 (Fig 7J). Upon binding with an alpha chain, CD18 is capable of forming a complex which plays a significant role in cellular adhesion and immune response [45]. Furthermore, deficiency of this molecule results in a phenotype similar to that described here, in vivo accumulation of HSCs in mouse BM [46]; in humans, down-regulation of CD18 expression was associated with a primitive HSC phenotype, and enhanced long-term repopulation in NSG mice [47]. To determine whether CD74 directly regulates CD18 expression, CD74 cytosolic domain binding to regulatory elements on CD18 chromatin was analyzed. As show in Fig 7K, the cytosolic domain of CD74 bound to the CD18 promoter area and to elements in the intron areas. Next, CD18 protein levels were determined in WT and CD74-deficient mice. A significant down-regulation in CD18 cell surface levels was observed on HSPCs and HSCs derived from mice lacking CD74 (Fig 7L and S6A Fig). Furthermore, reduced expression levels of CD18 were detected in the MIF$^{-/-}$ HSCs and HSPCs (Fig 7M and S6B Fig), further demonstrating that CD74 and its ligand, MIF, regulate CD18 expression.

Next, the effect of MIF inhibitor (ISO-1) on CD18 cell surface levels in HSPCs was analyzed. As shown in Fig 7N and S6C Fig, blocking MIF reduced CD18 cell surface expression on LSK- and CD34- cells. To further follow the role of extracellular MIF in the maintenance of HSPCS and CD18 regulation, WT HSPCs were cocultured with WT or MIF$^{-/-}$ BM cells. As shown in Fig 7O and S6D Fig, MIF secreted from the BM cells was essential for CD18 cell surface expression. In its absence, CD18 expression was reduced. Therefore, we suggest that MIF secreted from HSPCs or the BM environment binds to CD74 and induces CD18 expression; in the absence of MIF or CD74, its expression is down-regulated. To identify the β2 integrin complex regulated by CD74, we analyzed the α subunits that are affected by CD74 deficiency. As shown in the S6E Fig, CD74$^{-/-}$ HSPCs express lower levels of CD11B but not of CD11A or CD11C. The RNA-seq results (S5B Fig) support this finding (shown in the heatmap CD11B/

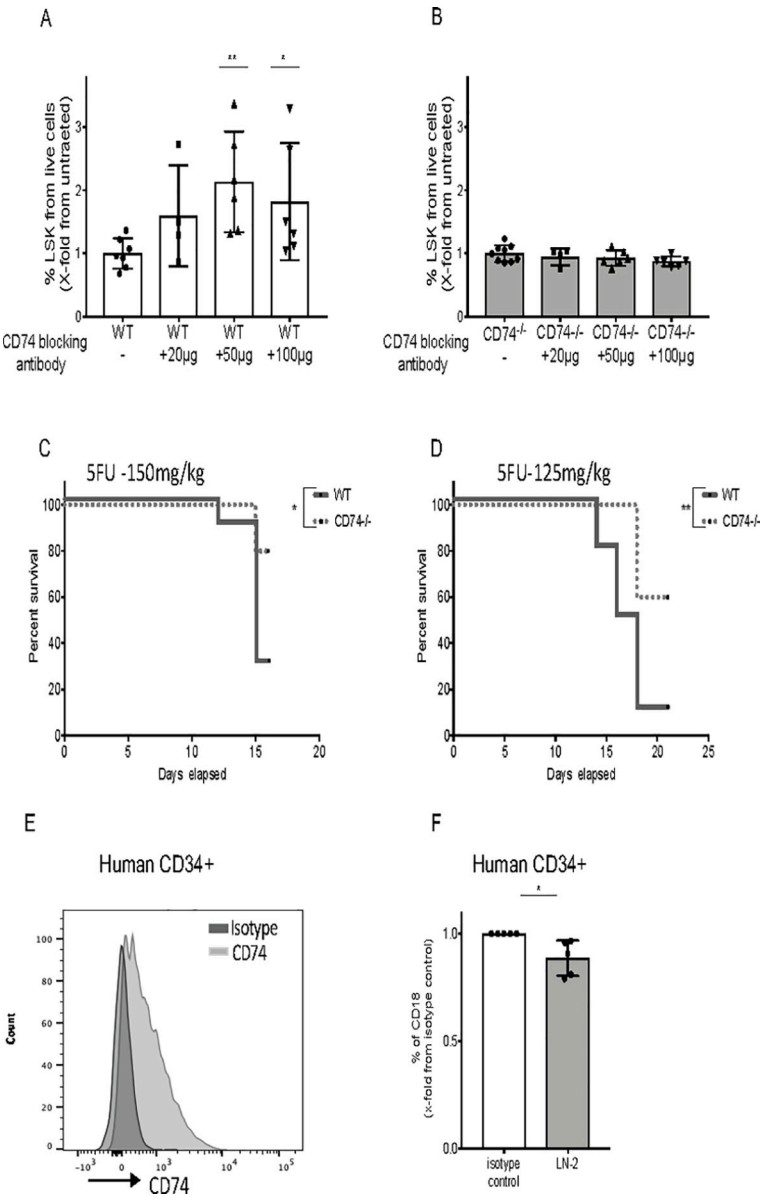

**Fig 8. CD74 can serve as a potential target for therapy. (A, B)** WT and CD74$^{-/-}$ BM cells were cultured alone or incubated with blocking anti-CD74 antibody (20, 50, and 100 µg/ml). After 48 h, percent LSK from live cells was analyzed by FACS; $n = 4$–7, Data A and B in S8 Data. **(C, D)** Survival curve: 5-FU (150 mg/kg and 125mg/kg) was injected to WT and CD74$^{-/-}$ mice once a week. Log-rank test $^{*}<0.05$; $n = 10$ in each group, Data C and D in S8 Data. **(G)** Human CD34+ BM cells were stained for CD74 cell surface expression by FACS; $n = 5$. **(H)** Human BM were cultured with anti-CD74 (LN2) antibody or isotype control for 48 h, and percent of CD18 on human CD34+ cells was analyzed by FACS; $n = 5$, Data E in S8 Data. Bars show SEM. Unpaired two-tailed t test $^{*}p < 0.05$; $^{**}p < 0.01$; $^{***}p < 0.001$. The fcs files and gates can be found at the Flow Repository (accession number FR-FCM-Z3F2). 5-FU, 5-fluorouracil; BM, bone marrow; FACS, fluorescence-activated cell sorting; WT, wild-type.

ITGAM). However, blocking CD74 activation by ISO-1 did not affect CD11B cell surface levels (S6F Fig), suggesting that CD74 does not directly regulate CD11B expression.

Finally, we analyzed the therapeutic potential of CD74 inhibition. First, we analyzed the effect of CD74 blockade on the ability of HSPCs to accumulate in vitro. Blocking CD74 using a specific antibody significantly elevated the percent and counts of LSK cells in WT mice, while

this antibody had no effect on CD74-deficient BM cells. Thus, CD74 has a direct role in the repopulation of the stem cell populations (Fig 8A and 8B, S7B Fig). The role of CD74 in hematological recovery from chemotherapy was followed. WT and CD74$^{-/-}$ mice were injected weekly with the cell cycle–dependent myelotoxic agent, 5-fluorouracil (5-FU), which kills proliferating cells and thereby stimulates HSC proliferation and the subsequent replenishment of the hematopoietic system [48]. Mice were injected with 150 mg/ml (Fig 8C) or 125 mg/ml (Fig 8D) 5-FU. After injection of the higher dose, at day 14, about 30% of the WT mice and about 80% of the CD74$^{-/-}$ mice survived. Injection of 125 mg/ml of the drug resulted in survival of about 10% WT and 60% CD74$^{-/-}$ mice at day 20. These results further suggest that elevated survival of CD74$^{-/-}$ HSCs allows better reconstitution of the immune system and increased survival of these mice under hemodepleting conditions. To verify the general nature of the phenomenon, CD74 expression was analyzed in human HSPCs (Gating strategy appears in S7A Fig). CD74 was detected on human CD34+ cells (Fig 8E), and its blocking resulted in down-regulation of CD18 expression (Fig 8F). These results suggest that CD74 regulates a similar pathway in human HSPCs and may provide a useful target for enhancing human stem cell transplantation.

## Discussion

Despite the enormous experience in the manipulation and therapeutic use of HSCs, the biology of these cells is still not fully understood. HSCs are useful cells for transplantation and for regenerative medicine. However, it has been not possible to date to expand adult HSCs without losing their self-renewal properties. We suggest that CD74 regulates HSPC maintenance, and its absence or inhibition can result in expansion of this highly potent population.

CD74 is a membrane-associated protein that serves as a cell surface receptor for the cytokine MIF [13]. Here we show that potent stem and progenitor cells are accumulated in the BM of CD74$^{-/-}$ mice, as well as in chimeric mice comprised of immune cells lacking CD74 and WT nonhematopoietic cells, suggesting that the phenotype of the donor cells and not the microenvironment is responsible for the enhanced CD74$^{-/-}$ HSC expansion. However, we cannot exclude the possibility that other immune cells contribute to the expansion of CD74$^{-/-}$ HSPCs in the chimeric mice. The increased quiescent stem cell population in mice lacking CD74 allows enhanced replenishment of the immune system and increased survival following chemotherapy. In the absence of CD74, there is an accumulation of higher numbers of HSPCs, which show a stronger potential to repopulate and compete for the BM niches compared to WT HSPCs. The accumulation does not result from induced stem cell homing or retention. Rather, our results show that the absence of CD74 leads to elevated survival of the HSCs. This results in elevated numbers of HSCs that express similar levels of ROS as wt HSPCs and proliferate at a similar rate. However, due to the elevated numbers of HSCs, the total production of ROS and the number of proliferating cells is increased. This allows the production of elevated numbers of HSPCs, resulting in generation of hematopoietic cells of the different lineages.

CD74 regulates CXCR4 expression [32]. It was recently shown that inhibition of the CXCR4/CXCL12 axis in the HSPC compartment results in an increase in mobilization efficiency. In addition, a concurrent increase in the cycling activity, self-renewing proliferation of the HSPC pool, and expansion of this population in the BM was observed [8]. Our results show that CD74 deficiency down-regulates CXCR4 cell surface expression. However, this regulation has a negligible role in the induced mobilization of CD74-deficient cells and does not affect cell proliferation, but rather supports survival of the cells. Nevertheless, it is possible that this lack of overall change might result from 2 antagonistic trends.

CD74 functions as a survival receptor in mature B cells [35,36], while results presented in this study show that the lack of CD74 leads to accumulation of functional HSPCs in the BM, due to their enhanced survival. The cell type-specific regulation of survival by CD74 might result from its association with different cell surface receptors resulting in induction of cell type-specific signals. Alternatively, it is possible that the environment of these 2 cell types controls the different outcomes. While the blood and spleen environment is normoxic, the BM environment is hypoxic. Niches of various stem cell types, including hematopoietic cells, are microenvironments with low oxygen tension, ranging from 1% to 8% $O_2$ [37]. The hypoxic environment plays a critical role in the regulation of stem cell self-renewal and differentiation [37]. Our results show that the hypoxic microenvironment plays a role in the CD74-induced outcome, providing an advantage for the CD74-deficient HSPCs. Moreover, CD74-deficient CD34- cells express lower levels of HIF-1α protein, suggesting a role for CD74 in the control of expression of this transcription factor, which is a key regulator in the adaptation to hypoxic conditions. Down-regulation of HIF-1α expression was previously shown to inhibit precursor cell differentiation [49,50]. We therefore suggest that the lower levels of HIF-1α might enable the accumulation of more primitive cells.

We suggest that, under normal conditions, CD74 fine-tunes the maintenance of the BM stem cells, inducing a cascade that leads to cell death. It was previously shown that mesenchymal stem cells produce MIF to delay hypoxia-induced senescence [51]. In addition, MIF protects against hypoxia/serum deprivation-induced apoptosis of mesenchymal stem cells by interacting with CD74 to stimulate c-Met, leading to downstream PI3K/Akt-FOXO3a signaling and decreased oxidative stress [52]. Thus, CD74 induces a cell type-specific cascade that leads to different cell death outcomes.

Our results show that CD18 is a target gene of CD74. We demonstrate that CD74-ICD binds to the promoter area and elements in the intron areas of CD18, inducing its transcription and expression. CD18 deficiency affects the expansion of HSCs in the murine BM [46]. In humans, down-regulation of CD18 expression is associated with a primitive HSC phenotype and enhanced long-term repopulating capacity in NSG mice [47]. Thus, CD74 and CD18 deficiency results in a similar HSPCs phenotype. This accumulation does not result from enhanced homing of cells to the BM, but, rather, is caused by a progressive cell-autonomous expansion of the cells. Thus, we can suggest that reduced expression levels of CD74 and its ligand MIF down-regulates CD18 expression, resulting in enhanced stem cell survival.

Several novel HSC transplantation protocols are based on the administration of high numbers of donor cells. The need for large numbers of engraftable cells becomes particularly challenging in the case of cord blood (CB) transplantation and adult HSC gene therapy protocols, because of suboptimal HSC doses available for infusion, and impaired engraftment of the transplanted cells [53]. Furthermore, in patients without a suitable donor, the HLA barrier can be overcome by transplantation of megadoses of highly purified mismatched CD34+ stem cells [54]. Methods to improve maintenance and expansion of HSPCs resulting in increased numbers of CB stem cells may shorten time to engraftment and improve survival in adult recipients. CD74 deficiency results in higher numbers of efficient stem cells, better reconstitution of the immune system, and increased survival of these mice under hemodepleting conditions.

We therefore suggest that in vitro blocking of CD74 expressed on HSPCs or its ligand MIF may lead to expansion of the stem cells and will improve transplantation protocols.

## Materials and methods

### Mice

C57BL/6 [17], CD74$^{-/-}$ [17], MIF$^{-/-}$ [55], and CD45.1 mice were used in this study. All mice were used at 6 to 8 weeks of age. To generate WT and CD74$^{-/-}$ littermates, C57BL/6 and CD74$^{-/-}$ mice were crossed, and screened for CD74$^{-/-}$ genotype by PCR.

NAC was administered by IP injection (50 mg kg$^{-1}$; Sigma, Germany) for 6 consecutive days before the experiments. 5-FU was administered by IP injection (150 mg kg$^{-1}$ or 125 mg kg$^{-1}$; ABIC, Teva Group, the Netherlands) once a week. AMD3100 was administered by IP injection (20 mg kg$^{-1}$; Sigma, Germany). The Weizmann Institute Animal Care and Use Committee approved all animal experiments. IACUC numbers: 2016–2019: 27730616–2; 2019–2022: 15340619–2.

### Cells

**Mice.**   BM cells were obtained by flushing long bones with PBS, and peripheral blood was collected from the eye or heart using heparinized syringes.

**Human samples.**   Human BM cells derived from healthy patients were provided by the hematology institute of the Kaplan Medical Center in compliance with the review board of the hospital. Consent was informed and the samples were obtained by written consent. The Weizmann institute review board approved the study: IRB number 1339–1.

### Library construction and sequencing

HSCs (CD34-/LSK) (1*10$^3$ cells) were sorted from WT and CD74$^{-/-}$ mice (BD FACS Aria III). Sequencing libraries were prepared using the SMART-Seq v4 Ultra Low Input RNA kit (Clonetech, United States of America). Sequencing libraries were constructed containing barcodes. Single-end reads were sequenced on one lane of an Illumina HiSeq2500 machine.

Sequence data analysis: Poly-A/T stretches and Illumina adapters were trimmed from the reads using cutadapt; resulting reads shorter than 30 bp were discarded. Reads were mapped to the *M. musculus* (GRCm38) reference genome using STAR [56], supplied with gene annotations downloaded from Ensembl release 92 (with EndToEnd option). Expression levels for each gene were quantified using htseq-count [57]. Differentially expressed genes were identified using DESeq2 (version 1.10.1) [58] with the betaPrior, cooksCutoff, and independentFiltering parameters set to False. Raw *P* values were adjusted for multiple testing using the procedure of Benjamini and Hochberg. The RNA-Seq data discussed in this publication have been deposited in NCBI's Gene Expression Omnibus [59] and are accessible through GEO Series accession number GSE163661 (https://www.ncbi.nlm.nih.gov/geo/query/acc.cgi?acc= GSE163661).

### Flow cytometry

For flow cytometry analysis, the following monoclonal antibodies were used: anti-mouse lineage cocktail (cat: 13302), CD11b (clone: M1/70), Gr-1 (clone: RB6-8C5), Ter119 (clone: TER-119), CD3 (clone: 145-2C11), B220 (clone: RA3-6B2), c-Kit (clone: 2B8), IgM (clone: RMM-1), IgD (clone: 11-26C), CD115 (clone: AFS98), CD150 (clone: TC15-12F12.2), CD48 (clone: HM48-1), CD45.1 (clone: A20), CD45.2 (clone: 104), CD18 (clone: M18/2) anti Rabbit (clone: poly4064), CD11A (clone: M17/4), CD11C (clone: N418), and CD44 (clone:IM7). For human staining: CD19 (clone: HIB19), CD3 (clone: VCHT1), CD8 (HIT8A), CD16 (3G8), CD38 (HIT2), and CD18 (clone:TS1/18), all from Biolegend, USA. Anti-Sca-1 (clone: D7), anti-CD34 (clone: RAM34), and anti-human CD74 (clone: 5–329) were purchased from

eBioscience, USA. Anti-CD74 (cat: FAB7478A) was purchased from R & D Systems, USA, and anti-CXCR4 (cat: TP-503, Torrey Pines Biolabs, USA). To analyze Hif-1α and MIF, total BM cells were stained for the designated markers (lineage/Sca-1/c-Kit/CD34), fixed and permeabilized using (cat: 00-5523-00, Invitrogen, United Kingdom), kit and stained with the Hif-1α antibody (cat: IC1935P; R&D) or MIF antibody (Abcam ab175189, UK). Human anti-CD34 (8G12) was obtained from BD Bioscience, USA. All analyses were done using a FACS Canto II flow cytometer (BD Bioscience, USA).

Sorting of the LSK CD45.1 WT and CD45.2 CD74$^{-/-}$ cells was performed using a FACS Aria II system (BD Bioscience, USA), following enrichment using CD117 (c-Kit) MicroBeads (cat: 130-091-224) on LS MACS Separation Columns (cat: 130-042-401), both obtained from Miltenyi Biotec, UK.

## Generation of chimeric mice

For the microenvironment experiment: Lethally irradiated (950 Rad) C57BL/6 (WT) recipient mice were reconstituted with $5^*10^6$ WT or CD74$^{-/-}$ BM cells. Additionally, lethally irradiated (950 rad) CD74$^{-/-}$ recipient mice on a C57BL/6 background were reconstituted with $5^*10^6$ WT or CD74$^{-/-}$ total BM cells. Long-term reconstitution of the peripheral blood and BM was evaluated at 16 weeks posttransplant.

Short-term chimera: Lethally irradiated (950 Rad) CD45.1 recipient mice were reconstituted with $25^*10^6$ of either WT(CD45.2) or CD74$^{-/-}$(CD45.2) BM cells. HSPCs in the BM were evaluated at 24 h, 72 h, and 1 week posttransplant.

**Competitive total BM transplant:** Lethally irradiated (950 Rad) WT recipient mice (CD45.1 on a C57BL/6 background) were reconstituted with $2.5 \times 10^6$ WT CD45.1 total BM cells together with either $2.5 \times 10^6$ CD45.2 WT or $2.5 \times 10^6$ CD45.2 CD74$^{-/-}$ total BM cells (1:1 ratio), $1.5 \times 10^6$ WT CD45.1 together with $0.5 \times 10^6$ CD45.2 WT or $0.5 \times 10^6$ CD45.2 CD74$^{-/-}$ (3:1 ratio), $1.75 \times 10^6$ WT CD45.1 together with $0.25 \times 10^6$ CD45.2 WT or $0.25 \times 10^6$ CD45.2 CD74$^{-/-}$ (7:1 ratio), or $1.8 \times 10^6$ WT CD45.1, together with $0.2 \times 10^6$ CD45.2 WT or $0.2 \times 10^6$ CD45.2 CD74$^{-/-}$ (9:1 ratio). Short- and long-term donor reconstitution (CD45.1 and CD45.2) was monitored 6, 16, and 24 weeks posttransplantation.

**Competitive LSK cell transplantation:** Lethally irradiated (950 Rad) WT recipient mice (CD45.1 on a C57BL/6 background) were reconstituted with $7.5^*10^4$ WT CD45.1 sorted LSK (Lin-/Sca-1+/c-Kit+) cells together with $7.5^*10^4$ CD45.2 CD74$^{-/-}$ sorted LSK (Lin-/Sca-1+/c-Kit+) cells.

## Colony-forming assay (CFU-C)

BM mononuclear (BM-MNC) cells were isolated by Ficoll separation and were seeded ($15 \times 10^3$ cells/ml) in CFU-C semisolid medium supplemented with EPO, IL-3, GM-CSF, and SCF, as described [60]. CFU-C were scored 7 days after plating, and results are presented as CFU-C per number of seeded cells.

## Serial transplantation

For serial transplantation assay, $2 \times 10^6$ BM cells were obtained from 6 WT donors and 6 CD74$^{-/-}$ donor mice and transplanted to lethally irradiated WT CD45.1 animals. Each donor population was transplanted to 4 to 5 recipient mice. At 10 to 12 weeks posttransplantation, 1 mouse from each donor served as a donor for the subsequent transplant.

## Cell cycle analysis

To analyze quiescent cells, total BM cells were stained for the designated markers (lineage/Sca-1/c-Kit/CD34), fixed and permeabilized using BD Cytofix/Cytoperm Plus kit (BD Bioscience, USA), and stained with the Ki-67 antibody (cat: 556026; BD Pharmingen, USA). To measure proliferation, C57BL/6 and CD74$^{-/-}$ mice were fed with 0.8 mg/ml BrdU in their drinking water for 3 days. BrdU incorporation was followed in LSK cells from BM using the BrdU flow kit (BD Pharmingen).

## ROS analysis

Cellular ROS levels were analyzed by incubation of BM cells with 2 μM hydroethidine (Molecular Probes, USA) for 10 min at 37˚C. Cells were then washed with PBS and stained for lineage/Sca-1/c-Kit/CD34 markers and analyzed by FACS.

## Apoptosis analysis

Total BM cells were stained with the appropriate antibodies, using the Annexin V binding buffer (1:10 in ddW; cat: 556454, BD Pharmingen, USA) and mixed with Annexin V (FITC Annexin V, cat: 556419; BD Pharmingen, USA). All samples were incubated for 15 min at room temperature. For measuring Annexin levels under hypoxia, the BM cells were incubated in a hypoxia chamber at 1% $O_2$ for 24 h before staining.

## Cell cultures

Coculture with Stroma: WT and CD74$^{-/-}$ BM cells ($2^*10^6$) were incubated with or without $10^5$ stroma cells (M210B4). After 48 h, percent LSK from total live cells was analyzed. Culture with Cytokines: WT and CD74$^{-/-}$ BM cells ($2^*10^6$) were incubated with or without 50 μg/ml stem cell factor (SCF) (cat#: 250–03; PEPROTECH, USA) and 50 μg/ml thrombopietin (TPO) (cat#315–14; PEPROPTECH, USA). After 48 h, percent LSK from live cells was analyzed.

For coculture of Lin negative cells with WT or MIF$^{-/-}$ BM cells: Lin negative cells were purified using EasySep mouse hematopoietic progenitor cell isolation kit (cat#19816A; STEMCELL TECHNOLOGIES, Canada). Then, $2^*10^5$ Lin negative cells (WT CD45.1) were incubated with $5^*10^6$ WT(CD45.2) or MIF$^{-/-}$ (CD45.2) BM in 1 media + 1% FCS for 48 h. Culture with the MIF inhibitor, ISO-1: WT BM cells ($5^*10^{6/}1$ ml in 1% FCS in media) were incubated with or without ISO-1 (ab142140; Abcam) for 48 h.

## ChIP-seq

WT BM lineage negative cells were purified using Lineage Cell Depletion Kit (Miltenyi cat#: 130-090-858). ChIP-seq was performed on 3 million cells, as described previously [32].

## Blocking of CD74

Mouse-cell culture with neutralizing anti-CD74 antibody: WT and CD74$^{-/-}$ BM cells ($10^{6/}$in 200 μl media) were incubated with or without neutralizing anti-CD74 antibody (YN-1) at 20, 50, and 100 μg/ml.

Human-BM cells from patients ($10^6$/in 200 μl) were treated for 48 h with isotype control or with anti-human CD74 (LN2; Biolegend) at 3 μg/ml.

## Supporting information

**S1 Fig.** **(A–D)** Gating strategy of HSPCs. **(E)** Sorted CD34-/LSK cells were analyzed for CD74 mRNA levels in WT and CD74$^{-/-}$ mice; $n = 3$. Bars show the DESeq2 normalized counts for the CD74 gene, Data A in S9 Data. **(F)** Lin- populations from WT and CD74$^{-/-}$ mice were analyzed for cKit and Sca-1 expression by FACS; $n = 15$, Data B in S9 Data. **(G)** C57BL/6 and CD74$^{-/-}$ mice were crossed to obtain WT and CD74$^{-/-}$ littermate mice. Percent of CD34-/LSK from WT and CD74$^{-/-}$ littermates; $n = 3$, Data C in S9 Data. **(H)** Sorted CD34-/LSK cells were analyzed for MIF mRNA levels in WT and CD74$^{-/-}$ mice; $n = 3$. Bars show the DESeq2 normalized counts for the MIF gene, Data D in S9 Data. **(I)** CD34-/LSK population from WT and CD74$^{-/-}$ mice was analyzed for MIF intracellular expression by FACS; $n = 13$, Data E in S9 Data. **(J, K)** Sorted CD34-/LSK cells were analyzed for CD44 mRNA levels in WT and CD74$^{-/-}$ mice; $n = 3$. Bars show the DESeq2 normalized counts for the CD44 gene, Data F in S9 Data **(J)** and cell surface CD44 expression; $n = 6$ **(K)**, Data G in S9 Data. Results are presented as mean −+ SD (unpaired two-tailed $t$ test $^*<0.05$ $^{**}$). The fcs files and gates can be found in FR-FCM-Z3F2. FACS, fluorescence-activated cell sorting; HSPC, hematopoietic stem and progenitor cell; MIF, migration inhibitory factor; WT, wild-type.
(PPTX)

**S2 Fig. MIF-deficient mice exhibit an accumulation of HSPCs.** BM cells derived from WT or MIF$^{-/-}$ were purified. **(A)** Total BM cellularity per femur and tibia, Data A in S10 Data **(B–D)** The percent of the different populations in WT and MIF$^{-/-}$-derived BM cells. **(B)** Lin-; Data B in S10 Data **(C)** LSK; Data C in S10 Data **(D)** CD34-; Data D in S10 Data $n = 8–9$. Results are presented as mean −+ SD (unpaired two-tailed $t$ test $^*<0.05$ $^{**}<0.005$ $^{***}<0.0005$). BM, bone marrow; HSPC, hematopoietic stem and progenitor cell; MIF, migration inhibitory factor; WT, wild-type.
(PPTX)

**S3 Fig.** **(A–H)** WT (CD45.1):WT (CD4.2) chimeric mice maintain a 1:1 ratio of donor cells. Lethally irradiated WT (CD45.1) mice were transplanted with BM derived from WT (CD45.2) mice at a 1:1 ratio. Percent of each population in the BM and PB was analyzed after 6, 16, and 24 weeks. **(A)** Total BM cells; Data A in S11 Data **(B)** BM myeloid cells; Data B in S11 Data **(C)** immature BM B cells; Data C in S11 Data **(D)** mature BM B cells; Data D in S11 Data **(E)** BM LSK; Data E in S11 Data **(F)** PB myeloid cells; Data F in S11 Data **(G)** PB mature B cells; Data G in S11 Data **(H)** PB LSK, and 4 h after injection of AMD3100; Data H in S11 Data **(I–L)** Lethally irradiated WT (CD45.1) mice were transplanted with BM derived from WT (CD45.1) and CD74$^{-/-}$ (CD45.2) mice at a 1:1 ratio. Percent of donor-derived cells was analyzed in the PB after 6, 16, and 24 weeks in **(I)** myeloid cells; Data I in S11 Data **(J)** immature BM B cells; Data J in S11 Data **(K)** PB LSK and 4 h after injection of AMD3100; Data K in S11 Data **(L)** PB mature B cells; Data L in S11 Data, $n = 6–27$. Bars show SEM. Unpaired two-tailed $t$ test $^*p < 0.05$; $^{**}p < 0.01$; $^{***}p < 0.001$; $^{****}p < 0.0001$. BM, bone marrow; PB, peripheral blood; WT, wild-type.
(PPTX)

**S4 Fig.** WT and CD74$^{-/-}$ HSPCs home similarly to the BM. **(A–E)** Lethally irradiated CD45.1 recipient mice were reconstituted with 25*10$^6$ of either WT (CD45.2) or CD74$^{-/-}$ (CD45.2) BM cells. Percent of Lin- and HSPCs in the BM was evaluated at 24 h **(A)** Data A in S12 Data, 72 h **(B, C)** Data B and C in S12 Data and 1 week **(D, E)**, Data D and E in S12 data, posttransplant; $n = 4–7$. **(F)** Lethally irradiated WT (CD45.1) mice were transplanted with BM derived from WT (CD45.1) and WT (CD45.2) at a 3:1 ratio, or BM derived from WT (CD45.1) and CD74$^{-/-}$ (CD45.2) mice at a 3:1 ratio. Mice were analyzed 16 weeks after transplantation. Dot

plots show the chimerism in the BM at the end of the experiment. **(G–L)** Lethally irradiated WT (CD45.1) mice were transplanted with BM derived from WT (CD45.1) and WT (CD45.2) at a 7:1 ratio, or BM derived from WT (CD45.1) and CD74$^{-/-}$ (CD45.2) mice at a 7:1 ratio. Mice were analyzed 16 weeks after transplantation. Percent of each population in the BM was analyzed 13 weeks; $n$ = 9–13. **(G)** Total BM cells; Data F in S12 Data **(H)** myeloid cells; Data G in S12 Data **(I)** immature BM B cells; Data H in S12 Data **(J)** LSK, Data I in S12 Data **(K)** CD34-/LSK, Data J in S12 Data **(L)** mature BM B cells, Data K in S12 Data. Bars show SEM. Unpaired two-tailed $t$ test $^*$<0.05 $^{**}$<0.01 $^{***}$<0.001 $^{****}$<0.0001. The fcs files and gates can be found in FR-FCM-Z3F2. BM, bone marrow; HSPC, hematopoietic stem and progenitor cell; WT, wild-type.
(PPTX)

**S5 Fig.** **(A)** Binding of CD74–ICD to CXCR4 promoter region in CLL cell samples. ChIP-seq analysis using anti-CD74 antibody. **(B)** HSCs (CD34-/LSK) ($10^3$ cells) were sorted from WT and CD74$^{-/-}$ mice. Differentially expressed genes were identified using DESeq2 (version 1.10.1); $n$ = 3. Analysis of the pathway was performed using Enrichr, a comprehensive gene set enrichment analysis. ChIP-seq analysis, chromatin immunoprecipitation-sequencing; HSC, hematopoietic stem cell; WT, wild-type.
(PPTX)

**S6 Fig. MIF/CD74 axis regulates CD18 expression.** **(A)** FACS analysis of LSK from WT and CD74$^{-/-}$ mice for CD18; $n$ = 6, Data A in S13. **(B)** FACS analysis of LSK from WT and MIF$^{-/-}$ mice for CD18; $n$ = 3, Data B in S13 Data. **(C)** WT and CD74$^{-/-}$ BM were cultured with MIF inhibitor (ISO-1) for 48 h, and percent CD18 on LSK was analyzed; $n$ = 6, Data C in S13 Data **(D)** WT (CD45.1) Lin negative cells were cultured in the presence of WT (CD45.2) total BM or MIF$^{-/-}$ (CD45.2) total BM for 48 h, the percent CD18 on LSK cells (CD45.1) was analyzed by FACS; $n$ = 8, Data D in S13 Data **(E)** LSK and CD34-/LSK populations from WT and CD74$^{-/-}$ mice were analyzed for cell surface expression of CD11A, CD11B, and CD11C; $n$ = 6 by FACS. Data E in S13 Data. **(F)** WT and CD74$^{-/-}$ BM were cultured with the MIF inhibitor (ISO-1) for 48 h, and percent CD11B on LSK and CD34-/LSK was analyzed by FACS; $n$ = 4, Data F in S13 Data. Bars show SEM. Unpaired two-tailed $t$ test $^*p < 0.05$; $^{**}p < 0.01$; $^{***}p < 0.001$; $^{****}p < 0.0001$. The fcs files and gates can be found in FR-FCM-Z3F2. BM, bone marrow; FACS, fluorescence-activated cell sorting; MIF, migration inhibitory factor; WT, wild-type.
(PPTX)

**S7 Fig.** **(A)** Gating strategy for human CD34+ cells. **(B)** WT and CD74$^{-/-}$ BM cells were cultured alone or incubated with blocking anti-CD74 antibody (20, 50, and 100 mg/ml). After 48 h, LSK expressing cells from 200,000 cells were analyzed by FACS; $n$ = 4–7, Data A in S14 Data. The underlying numerical data for this figure can be found in S14 Data, and fcs files and gates can be found in FR-FCM-Z3F2. BM, bone marrow; FACS, fluorescence-activated cell sorting; WT, wild-type.
(PPTX)

**S1 Data. Values for each data point to create the graphs in Fig 1.**
(XLSX)

**S2 Data. Values for each data point to create the graphs in Fig 2.**
(XLSX)

**S3 Data. Values for each data point to create the graphs in Fig 3.**
(XLSX)

**S4 Data. Values for each data point to create the graphs in Fig 4.**
(XLSX)

**S5 Data. Values for each data point to create the graphs in Fig 5.**
(XLSX)

**S6 Data. Values for each data point to create the graphs in Fig 6.**
(XLSX)

**S7 Data. Values for each data point to create the graphs in Fig 7.**
(XLSX)

**S8 Data. Values for each data point to create the graphs in Fig 8.**
(XLSX)

**S9 Data. Values for each data point to create the graphs in S1 Fig.**
(XLSX)

**S10 Data. Values for each data point to create the graphs in S2 Fig.**
(XLSX)

**S11 Data. Values for each data point to create the graphs in S3 Fig.**
(XLSX)

**S12 Data. Values for each data point to create the graphs in S4 Fig.**
(XLSX)

**S13 Data. Values for each data point to create the graphs in S6 Fig.**
(XLSX)

**S14 Data. Values for each data point to create the graphs in S7 Fig.**
(XLSX)

## Acknowledgments

The authors wish to thank members of the Shachar lab for fruitful discussion and support, and Gilgi Friedlander at the Nancy and Stephen Grand Israel National Center for Personalized Medicine. I.S. is the incumbent of the Dr. Morton and Ann Kleiman Professorial Chair. G.F. is the Incumbent of the David and Stacey Cynamon Research fellow Chair in Genetics and Personalized Medicine.

## Author Contributions

**Conceptualization:** Shirly Becker-Herman, Lital Sever, Richard Bucala, Amnon Peled, Idit Shachar.

**Data curation:** Shirly Becker-Herman, Milena Rozenberg, Carmit Hillel-Karniel, Naama Gil-Yarom, Mattias P. Kramer, Avital Barak, Keren David, Lihi Radomir, Hadas Lewinsky, Michal Levi.

**Formal analysis:** Shirly Becker-Herman, Milena Rozenberg, Carmit Hillel-Karniel, Naama Gil-Yarom, Mattias P. Kramer, Avital Barak, Lital Sever, Keren David, Lihi Radomir, Hadas Lewinsky, Michal Levi, Gilgi Friedlander.

**Funding acquisition:** Idit Shachar.

**Investigation:** Shirly Becker-Herman, Idit Shachar.

**Resources:** Richard Bucala.

**Supervision:** Idit Shachar.

**Validation:** Shirly Becker-Herman.

**Writing – original draft:** Shirly Becker-Herman.

**Writing – review & editing:** Richard Bucala, Amnon Peled, Idit Shachar.

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
