## [Editor Report · Decision Letter 0]

9 Apr 2020

Dear Dr Shachar, 

Thank you for submitting your manuscript entitled "CD74 is a regulator of hematopoietic stem cell maintenance" for consideration as a Research Article by PLOS Biology.

Your manuscript has now been evaluated by the PLOS Biology editorial staff as well as by an academic editor with relevant expertise and I am writing to let you know that we would like to send your submission out for external peer review.

Please re-submit your manuscript within two working days, i.e. by Apr 11 2020 11:59PM.

Kind regards,

Di Jiang,

Associate Editor

PLOS Biology

---

## [Decision Letter · Decision Letter 1]

30 Apr 2020

Dear Dr Shachar,

Thank you very much for submitting your manuscript "CD74 is a regulator of hematopoietic stem cell maintenance" for consideration as a Research Article at PLOS Biology. Your manuscript has been evaluated by the PLOS Biology editors, an Academic Editor with relevant expertise, and by two independent reviewers.

In light of the reviews (below), we will welcome re-submission of a revised version that takes into account the reviewers' comments. We cannot make any decision about publication until we have seen the revised manuscript and your response to the reviewers' comments. Your revised manuscript is also likely to be sent for further evaluation by the reviewers.

We expect to receive your revised manuscript within 2 months. Please note given we are in the midst of the COVID-19 pandemic, we are flexible regarding turnaround time for revision.

**IMPORTANT - SUBMITTING YOUR REVISION**

*Re-submission Checklist*

*Published Peer Review*

*PLOS Data Policy*

*Blot and Gel Data Policy*

Sincerely,

Di Jiang, PhD

Associate Editor

PLOS Biology

REVIEWS:

Reviewer #1: This paper shows the role of CD74, a nonpolymorphic type II integral membrane　protein (Stumptner-Cuvelette and Benaroch, Biochim Biophys Acta, 2002) is a member of the regulated intramembrane proteolysis-processed protein family (Becker-Herman S et al, Mol Biol Cell, 2005) in regulating the HSPC (hematopoietic stem cell progenitor cells) maintenance.

Becker-Herman et al analysed the CD74-deficient mice and they demonstrate expansion of HSPC populations in CD74-/- mice, displaying enhanced long-term self-renewal capacity and high CD74 HSC survival. 

Major comments:

1) The main novelty of this paper is the finding that the CD74 deficiency results in expansion of the highly potent HSC population. Regarding the molecular mechanism, this paper has focused on retention of HSCs and HSPCs. It would be very nice, if they could see how CD74 molecules works for HSC retention with CD18.

2) According to the result of the effect of CD74 blockade and the role of CD74 in haematological recovery from chemotherapy with 5-FU treatment, it helps improve clinical insight into bone marrow transplant protocols and enable improved engraftment. You should discuss about the more practical translation in the discussion section. 

3) The experiments are well-designed, while it may be more convincible if comprehensive analysis is explored. Such as, BM populations of the mixed chimeras at various time points are showed in both WT CD45.2 VS. WT CD45.1 and CD74-/- (CD45.2) VS. WT CD45.1. How about PB populations of the mixed chimeras at various time in WT CD45.2 VS. WT CD45.1 and the total PB and LSK in CD74-/- (CD45.2) VS. WT CD45.1? (Figure 3 and Supplementary Figure 3.) And also, what's the additional findings, like survival-related pathways or other genes except CD18, are affected by CD74-/- in your RNA-seq data? (Figure 7-J)

4) Most conclusions are well-supported by the data and well-designed experiment. However, there are few conclusions maybe need further discuss. For example, In Figure 2 A-C, the ratio of % from total BM cells in LIN-, LSK, and CD34-/LSK subpopulation between CD74-/- BM and WT donors is around 1-3. In Figure 2 E-F, the ratio % LSK from total BM between CD74-/- and WT stem cell numbers under cellular microenvironment (stroma, cytokines (TPO and SCF)) is also around 1-3. 

5) They have mentioned "These conditions (cellular microenvironment) had only a minor or no effect on the fold of increase in WT and CD74-/- stem cell numbers" and draw the conclusion "the accumulation potential of HSPCs in the absence of CD74 is intrinsic to the cells."

 How about the fold change in LIN- and CD34-/LSK between CD74-/- and WT? It may be more clear if they show the fold change between CD74-/- and WT within the same condition and put the fold change number in Figure 2 A-C as well, and also in Figure 5. When check the H-K, in the WT:CD74-/- chimera, the numbers of BM Myeloid, immature, LSK and CD34-/LSK is different, especially CD34-/LSK. It is difficult to be convinced when describe them "equal" in the paper. How about adjust to 5:1 dilution or 9:1 and then check the difference?

6) In their previous paper (Gore Y et al, J Biol Chem, 2008), it shows that MIF binds to a complex of CD74 and CD44, resulting in initiation of a signaling pathway. They should discuss about the role of CD44 and the change of the signaling cascade. 

Minor comments:

1) Figure 1-B, -I, -J, -K, Figure 2-C, Figure 5-G, -H, -I, -J, -K, Figure 6-D, -E, G, Figure 7-A, -B, Figure 8-A and also Supplementary Figure 4-C and -E, the error bars are relatively high. May I ask what's your interpretation?

2) Spelling mistakes. e.g. 'may be' (Page 6), 'TPO' (instead 'EPO') (Page 24, line 2), 'WT (CD45.2)' (Supplementary Figure 3), 'WT (CD45.1)' (Figure 3-A).

3) The format. e.g. '2Internal' (Page 8, line 5), the sentence should be bold (Page 12, line 1 and Supplementary Figure 5), divide the subtitle and the paragraph (Page 23, line 18 and line 24), the arrow in the figure. (Figure 1- A and D, Supplementary Figure 1 A-D, Figure 3, Figure 6, Figure 7, Supplementary Figure 4 and Supplementary Figure 6.)

4) The font. e.g. Lin- VS. Lin-, CD34+ VS. CD34+, CD34- VS. CD34-, Gated: LIN- VS. Gated:Lin-, CD74-/- VS. CD74-/-, MIF-/- VS. MIF-/-, Sca-1+ VS. Sca1+. (Figure 1, Supplementary Figure 1, Supplementary Figure 2, Page 12, line 3, line 13-14 and line 21.)

5) Without Space. e.g. '% from' (Figure 1-I), 'per 15000' (Figure 1-K), '25% WT' (Supplementary Figure 4-F).

6) Others. e.g. capital (Supplementary Figure 3), '0.25 x 106' (Page 23, line 21), grammar mistake (Page 17, line 23), etc.

Reviewer #2: 

The manuscript by Becker-Herman et al is a very interesting study demonstrating that lack of CD74 expression on hematopoietic cells has a hematopoietic-cell-autonomous effect resulting in increased HSPC survival and hematopoietic reconstitution after transplantation. The authors convincingly demonstrate that the effect is not dependent on CD74 expression in non-hematopoietic cells and is partly mimicked in MIF KO suggesting that MIF-CD74 negatively regulates HSPC survival, which interestingly is the opposite effect of this pathway on mature B cells. The authors propose several potential mechanisms which remain to be functionally tested, leaving good food for future studies. They go to the length of doing four rounds of serial bone marrow transplantations and limiting dilutions with different cell concentrations to study potential effects on HSC self-renewal, which is quite impressive. Finally they perform repeated 5-FU administrations and show increased survival of CD74 KO mice and suggest that blocking CD74 might serve to improve the outcome of clinical hematopoietic stem cell transplantations. The manuscript is very clearly written and easy to follow. I only have relatively minor comments aimed at improving clarity and perhaps a more balanced interpretation of the results.

Major comments:

1) The authors convincingly demonstrate that the CD74 deletion in non-hematopoietic cells does not contribute to the phenotype, but the experiments do not exclude the possibility that CD74 in other hematopoietic cells plays a role. This would fit with the kinetics since the effects take some time to manifest after transplantation - perhaps should be discussed.

2) The effects of CD74 are clearly cell-autonomous but still require MIF binding? A number of growth factors was used but was MIF not used for in vitro culture? Is MIF produced by HSCs and acting through an autocrine loop?

3) Please explain how you define the different cell populations (including mature and immature B cells) immunophenotypically in the figure legend.

4) The authors propose several potential mechanisms explaining the increased HSC survival in CD74 KO but no definitive functional explanation. Different survival effects on HSCs and B cells is very intriguing. The authors hypothesize that increased hypoxia in HSCs might explain the difference and look into Hif1a. If CD74 induces Hif1a transcription, why are CD74 KO cells more responsive and fit under hypoxia, which stabilizes Hif1a?

5) Since the last time point in Fig. 8C is very close to the point where the curves separate from each other it would be best to show longer evolution.

6) Have the authors tested human HSPCs (like cord-blood-derived, which typically show reduced engraftment) with the CD74 blocking antibody? Not essential but would substantiate the human relevance.

7) They show the frequency of LSK cells in BM cells treated in vitro with anti-CD74 Ab, but this could reflect effects on other cell populations - enumerating LSK cells would be better.

8) Could the blocking antibody be well tolerated in vivo or cause lymphopenia?

Minor comments:

P.6 1st para. Rephrase concluding remark to "MIF/CD74 limits HSPC numbers, as opposed to the prosurvival effect in B cells" or similar.

P. 6. 2nd para. Refine "Thus, the lack of CD74 in hematopoietic cells…"

T cells last longer but should be eventually replaced by donor-derived cells.

For the CXCR4 part, there seem to be two antagonistic changes which might explain unchanged mobilization by AMD3100 despite lower CXCR4 expression (higher number of BM HSPCs but lower CXCR4 expression) - perhaps deserves some comment in the discussion.

---

## [Editor Report · Decision Letter 2]

17 Dec 2020

Dear Dr Shachar,

Thank you for submitting your revised Research Article entitled "CD74 is a regulator of hematopoietic stem cell maintenance" for publication in PLOS Biology. I have now obtained advice from the Academic Editor and have discussed the revision and rebuttal with the team of editors. 

Based on these comments, we will probably accept this manuscript for publication, assuming that you will modify the manuscript to address the data and other policy-related requests noted at the end of this email.

We expect to receive your revised manuscript within two weeks. Your revisions should address the specific points made by each reviewer. 

-  a cover letter that should detail your responses to any editorial requests.

*Published Peer Review History*

*Early Version*

Sincerely,

Richard Hodge, PhD

Associate Editor,

rhodge@plos.org,

PLOS Biology

ETHICS STATEMENT:

Thank you for including the following ethics statements:

“The Weizmann Institute Animal Care and Use Committee approved all animal experiments."

"Human samples- BM cells derived from healthy patients was provided in compliance with the institutional review boards of the participating hospitals”

At this time, we ask that you please include the following details:

(a) Please amend your current ethics statement to include the full name of the institutional review boards that approved your specific study, including the names of the specific hospitals. We additionally ask that you include your IRB approval numbers in your ethics statement.

(b) Please provide additional details regarding participant consent for the use of bone marrow cells from healthy patients. In the ethics statement in the Methods and online submission information, please ensure that you have specified (1) whether consent was informed and (2) what type you obtained (for instance, written or verbal, and if verbal, how it was documented and witnessed). If your study included minors, state whether you obtained consent from parents or guardians. If the need for consent was waived by the ethics committee, please include this information. On the other hand, if the patient samples were accessed anonymously, please include this information. 

(c) Please include the approval number issued by the Weizmann IACUC for the animal studies conducted in your manuscript.

You may be aware of the PLOS Data Policy, which requires that all data be made publicly available without restriction before we can accept the manuscript: http://journals.plos.org/plosbiology/s/data-availability. For more information, please also see this editorial: http://dx.doi.org/10.1371/journal.pbio.1001797

Fig. 1B, C, E-K; Fig. 2A-F; Fig. 3B-H; Fig. 4A-E; Fig. 5A-P; Fig. 6B, D-K; Fig. 7A-G, I, J, L-O; Fig. 8A-F; Fig. S1E-K; Fig. S2A-D; Fig. S3A-L; Fig. S4A-E, G-L; Fig. S6A-F and Fig. S7B

For figures containing FACS data, we ask that you provide FCS files and a picture showing the successive plots and gates that were applied to the FCS files to generate the figure.

*Please also ensure that figure legends in your manuscript include information on WHERE THE UNDERLYING DATA CAN BE FOUND where the underlying data can be found, and ensure your supplemental data file/s has a legend*

Finally, the RNA-Sequencing data presented in the manuscript should be deposited in a publicly available database such as GEO. We ask that you please amend your Data Availability Statement to list the name of the repository where the RNA-Seq can be found and the accompanying accession numbers. Please see https://journals.plos.org/plosbiology/s/data-availability for additional information.

---

## [Editor Report · Decision Letter 3]

12 Jan 2021

Dear Dr Shachar,

Thank you for submitting your revised Research Article entitled "CD74 is a regulator of hematopoietic stem cell maintenance" for publication in PLOS Biology and for addressing our editorial requests. At this time, we ask that you please provide some additional information before your manuscript can be accepted for publication.

We expect to receive your revised manuscript within one week. Your revisions should address the specific points made by each reviewer.

To submit your revision, please go to https://www.editorialmanager.com/pbiology/ and log in as an Author.

Click the link labelled 'Submissions Needing Revision' to find your submission record. Your revised submission must include the following:

- a cover letter that should detail your responses to any editorial requests

(1) ETHICS STATEMENT:

Thank you for including your ethics statement:

‘Human BM cells derived from healthy patients were provided by the hematology institute of the Kaplan Medical Center in compliance with the review board of the hospital. The samples were obtained by written consent. IRB number 1339-1.’

(a) Please amend your current ethics statement to include the full name of the institutional review board that approved your specific study.

(b) Please amend your current ethics statement to state that the institutional review board specifically approved your study.

(c) Please provide additional details regarding participant consent. In the ethics statement, please ensure that you have specified that consent was informed.

(a) Thank you for depositing the RNA-Sequencing data in the GEO database and for providing the accession number. However, we note that GSE163661 is currently private and is scheduled to be released in December 2023. You should make this data publicly available now for us to be able to process your manuscript for Production.

(b) For figures containing FACS data, we ask that you please remove the external Dropbox files provided in your submission and submit the FCS files to the FlowRepository (https://flowrepository.org/) instead given the size of the files. As before, we ask that this data be made publicly available in the repository before publication.

(c) We note that your Data Availability statement currently states:

‘Data are from the Weizmann Institute study whose authors my bee contacted by email’

Please update this statement to ensure you have provided all necessary access information in complete sentences (including the raw data in the Supplementary Files provided in your submission, FACs files and RNA-Seq data deposited in the GEO database). This statement will appear on the published version of your manuscript.

Sincerely,

Richard Hodge, PhD

Associate Editor

PLOS Biology

---

## [Editor Report · Decision Letter 4]

18 Jan 2021

Dear Dr Shachar,

Thank you for submitting your revised Research Article after our e-mail communication entitled "CD74 is a regulator of hematopoietic stem cell maintenance" for publication in PLOS Biology and for addressing our editorial requests. Please accept my apologises, but during the editorial checks I have noticed during that we require some additional information before your manuscript can be accepted for publication. 

We expect to receive your revised manuscript within one week. Your revisions should address the specific points made by each reviewer.

To submit your revision, please go to https://www.editorialmanager.com/pbiology/ and log in as an Author.

Click the link labelled 'Submissions Needing Revision' to find your submission record. Your revised submission must include the following:

- a cover letter that should detail your responses to any editorial requests

1. We note that the following sentence is still present in your revised manuscript (page 20):

‘For original data, please contact idit.shachar@weizmann.ac.il’

At this time, we ask that you please delete this sentence in both the manuscript and in the Details page in Editorial Manager since you have provided all of the Supplementary Data in the manuscript or in the indicated data repositories.

2. Please update your Data Availability Statement in the manuscript so that the accession number of the data deposited in the Flow Repository is specifically stated. For example, please update your statement to:

‘All raw data for the Figures presented in the manuscript can be found in the Supplementary Information. The RNA-Seq data discussed in this publication have been deposited in NCBI's Gene Expression Omnibus and are accessible through GEO Series accession number GSE163661 (https://www.ncbi.nlm.nih.gov/geo/query/acc.cgi?acc= GSE163661). The FCS files for the flow cytometry data can be found in the FlowRepository (https://flowrepository.org/) with accession number…..’

We cannot accept your manuscript for publication without the Flow Repository accession number. 

3. Please update your Figure legends to specifically state that the FCS files for the flow cytometry can be found at the Flow Repository, *along with the accession number*, where appropriate. Please update your Figure legends in this way for any Figures that include flow cytometry data. 

For example, the Figure 1 legend currently states:

"The underlying numerical data for this figure can be found in S1_Data and fcs files and gates can be found in S1F_Data"

However, S1F_Data has been removed from the File Inventory since this data is now deposited at the Flow Repository. Please update this Figure legend, and other all other Figure legends that contain flow cytometry data, to state that the FCS files can be found at the Flow Repository (accession number xxxx).

Please do not hesitate to e-mail me directly at rhodge@plos.org if you have any questions or concerns. As soon as the points above have been addressed, we will be able to accept your manuscript for publication. Please accept my apologises for not noticing the flow cytometry data availability statements earlier and including this in our earlier e-mail correspondence. 

Sincerely,

Richard Hodge, PhD

Associate Editor

PLOS Biology

---

## [Editor Report · Decision Letter 5]

29 Jan 2021

Dear Dr Shachar,

On behalf of my colleagues and the Academic Editor, Connie Eaves, I am pleased to say that we can in principle offer to publish your Research Article "CD74 is a regulator of hematopoietic stem cell maintenance" in PLOS Biology, provided you address any remaining formatting and reporting issues. These will be detailed in an email that will follow this letter and that you will usually receive within 2-3 business days, during which time no action is required from you. Please note that we will not be able to formally accept your manuscript and schedule it for publication until you have made the required changes.

PRESS

Thank you again for supporting Open Access publishing. We look forward to publishing your paper in PLOS Biology. 

Kind regards,

Richard

Richard Hodge, PhD

Associate Editor, PLOS Biology

rhodge@plos.org